# NQO1 targeting prodrug triggers innate sensing to overcome checkpoint blockade resistance

Xiaoguang Li[1,2,5], Zhida Liu [1,5], Anli Zhang[1], Chuanhui Han[1], Aijun Shen[1], Lingxiang Jiang[3], David A. Boothman[4], Jian Qiao[1], Yang Wang[1], Xiumei Huang[3] & Yang-Xin Fu [1]

Lack of proper innate sensing inside tumor microenvironment (TME) limits T cell-targeted immunotherapy. NAD(P)H:quinone oxidoreductase 1 (NQO1) is highly enriched in multiple tumor types and has emerged as a promising target for direct tumor-killing. Here, we demonstrate that NQO1-targeting prodrug β-lapachone triggers tumor-selective innate sensing leading to T cell-dependent tumor control. β-Lapachone is catalyzed and bioactivated by NQO1 to generate ROS in NQO1[high] tumor cells triggering oxidative stress and release of the damage signals for innate sensing. β-Lapachone-induced high mobility group box 1 (HMGB1) release activates the host TLR4/MyD88/type I interferon pathway and Batf3 dendritic cell-dependent cross-priming to bridge innate and adaptive immune responses against the tumor. Furthermore, targeting NQO1 is very potent to trigger innate sensing for T cell re-activation to overcome checkpoint blockade resistance in well-established tumors. Our study reveals that targeting NQO1 potently triggers innate sensing within TME that synergizes with immunotherapy to overcome adaptive resistance.

[1] Department of Pathology, University of Texas Southwestern Medical Center, Dallas, TX 75235, USA. [2] School of Public Health, Shanghai Jiao Tong University School of Medicine, 200025 Shanghai, China. [3] Department of Radiation Oncology, Simon Cancer Center, Indiana University School of Medicine, Indianapolis, IN 46022, USA. [4] Department of Biochemistry and Molecular Biology, Simon Cancer Center, Indiana University School of Medicine, Indianapolis, IN 46202, USA. [5]These authors contributed equally: Xiaoguang Li, Zhida Liu. Correspondence and requests for materials should be addressed to X.H. (email: xiuhuang@iu.edu) or to Y.-X.F. (email: Yang-Xin.Fu@UTSouthwestern.edu)

Programmed death 1 (PD-1): Programmed Death-Ligand 1 (PD-L1) monoclonal antibodies have demonstrated efficacious and durable responses across several different cancer types, and these initial clinical successes have highlighted the field of cancer immunotherapy[1,2]. Unfortunately, only a minority of patients treated with anti-PD-1/PD-L1 agents have durable responses[3]. While other checkpoints may contribute to low response and relapse, additional checkpoint blockades on T cells have not improved response rates[4,5]. The dysfunctional antigen-presenting cells (APCs) inside tumor microenvironment might limit optimal activation of T cells[6]. While tumor antigens might be available inside tumor environment, lack of proper antigen processing and presentation might limit tumor-specific T cells to be reactivated.

The generation of effective adaptive immunity requires the coordinative innate immune response, including sensing of danger or damage signals to activate innate cells (i.e. dendritic cells, macrophages, and natural killer cells), antigen processing and presentation, type I IFN production, and cross-priming of T cells[7]. In general, these danger or damage signals are recognized by extracellular and intracellular pattern recognition receptors (PRRs) expressed by innate immune cells, and promote the uptake of antigens, activate APCs, and facilitate the interaction between APCs and damaged cells[8]. However, these critical properties of normal innate immune responses are often corrupted in the tumor microenvironment[9]. For example, cancers pervasively favor the survival of tumor clones lacking or unable to present adequate neo-antigens; tumors also prefer PRRs signals or dysfunctional innate immune cells that promote cancer inflammation rather than priming an adaptive response[6,10,11].

Since tumors exhibit impaired innate sensing that favors an immunosuppressive microenvironment, an important consideration in improving checkpoint blockade is to enhance innate signals in the tumor microenvironment. One possible approach to achieve this goal involves the induction of immunogenic cell death (ICD) within the tumor microenvironment[9]. For example, a number of DNA-damaging or DNA repair inhibiting chemotherapies (such as anthracyclines and oxaliplatin) and radiotherapy elicit ICD[12]. ICD is characterized by release of a series of immunostimulatory damage-associated molecular patterns (DAMPs) such as high mobility group box 1 (HMGB1) protein, extracellular ATP, cytoplasmic calreticulin, and endogenous nucleic acids by the dying tumor cells[13]. These DAMPs are recognized by their cognate PRRs expressed by innate immune cells. This DAMP/PRR signaling alters the inflammatory microenvironment and/or stimulates neoantigen production, and attracts and activates APCs to activate T cells, which are now licensed to attack the tumor[14]. Thus, the immunostimulatory properties make ICD-inducing agents attractive candidates for combining with immunotherapy. However, only a few cytotoxic drugs have been identified as ICDs inducers, and the general toxicities, immune suppressive nature, and lack of tumor selectivity limit their use. Thus, it is highly desirable to explore whether "targeted" agents can more specifically increase innate sensing and subsequently expand the benefits of anit-PD-1/PD-L1 treatment.

NAD(P)H:quinone oxidoreductase 1 (NQO1) is a cytosolic two-electron oxidoreductase which is upregulated in many human cancers[15,16]. High-level expression of NQO1 is associated with late clinical stage, poor prognosis and lymph node metastasis[17,18]. NQO1 bioactivatable drugs, including β-lapachone (β-lap, in clinical form, ARQ761), have a unique quinone structure which can be catalyzed by NQO1 to generate reactive oxygen species (ROS)[19]. In general, one mole of β-lap generates ~120 moles of superoxide, consuming ~60 moles of NAD(P)H in ~2 min[20]. NQO1 is overexpressed in tumor cells

and catalase, a hydrogen peroxide ($H_2O_2$) scavenging enzyme, is lost in tumor tissues versus normal tissue[21]. High NQO1:Catalase ratios in human cancers can offer an optimal therapeutic window for the use of NQO1 bioactivatable drugs, while low expression ratios protect normal tissues. The intensive tumor-specific ROS production leads to extensive oxidative DNA lesions and tumor selective cell death[22]. It has been demonstrated that NQO1 bioactivaible β-lap causes unrepaired DNA damage and cell death and synergizes with PARP1 inhibitors and radiotherapy in xenograft models[19,23]. β-Lap is currently being tested in monotherapy or in combination with the other chemodrugs in patients with NQO1+ solid tumors (ClinicalTrials.gov identifiers NCT02514031 and NCT01502800). However, evaluations of the antitumor efficacy of β-lap were mainly carried out in vitro and in immunodeficient mouse models, and improving therapy often focused on enhanced direct tumor killing with little attention to adaptive immunity.

Here we screen and identify high NQO1 expression in murine tumor lines. Using syngeneic or T cell reconstituted human xenograft mouse models, we report that β-lap-induced tumor control largely depends on adaptive immunity through tumor-specific innate sensing for adaptive immunity leading to tumor control. Furthermore, targeting NQO1 is very potent to trigger innate sensing and overcome checkpoint blockade resistance.

## Results

**β-Lap selectively suppresses NQO1+ murine tumor growth.** For years, the evaluations of the antitumor efficacy of β-lap have been mainly carried out direct killing assay by in vitro assays or in immunodeficient mouse models. To study whether this prodrug can trigger immune responses, multiple murine cancer cell lines were screened to examine the role of NQO1 in β-lap function. Tumor cell lines (MC38 colorectal adenocarcinoma, TC-1 lung cancer and Ag104Ld fibrosarcoma) that express a high level of NQO1 (Supplementary Fig. 1a) were sensitive to β-lap exposure (Fig. 1a). In contrast, NQO1-deficient cell lines, B16 (melanoma) and Panc02 (pancreatic cancer) (Supplementary Fig. 1a), were resistant to β-lap exposure (Fig. 1a). Dicoumarol, an NQO1 specific inhibitor, reversed the NQO1-mediated lethality (Fig. 1b). Next, we determined whether depletion of NQO1 abrogates the cytotoxicity of β-lap. CRISPR-mediated NQO1 knockout (Supplementary Fig. 1b) endowed MC38 cells resistance to β-lap treatment (Fig. 1c, Supplementary Fig. 1c). Similarly, overexpression of NQO1 in B16 cells (Supplementary Fig. 1d) led to sensitivity to β-lap (Fig. 1d, Supplementary Fig. 1e), and inhibition of NQO1 by dicoumarol spared β-lap lethality (Supplementary Fig. 1f). Lethal dose of β-lap caused rapid cell swelling, membrane rupture, and Annexin V+/7AAD+ cell death (Fig. 1e). NQO1 catalyzes the two-electron oxidoreduction of β-lap to generate high levels of ROS (i.e. hydrogen peroxide/$H_2O_2$), causing massive DNA oxidation and cell death[19]. Indeed, β-lap induced high level of ROS in NQO1-positive murine tumor lines and much less in NQO1-null lines (Fig. 1f, g). Inhibition of NQO1 by dicoumarol abolished this effect (Fig. 1f, g). Next, we determined whether neutralizing ROS could inhibit β-lap-induced cell lethality. Catalase, an $H_2O_2$ scavenging enzyme, significantly spared β-lap-mediated lethality (Fig. 1h, Supplementary Fig. 1g). These results suggest that β-lap induces NQO1+ tumor cell death through intensive tumor-specific ROS production in vitro.

We further examined the antitumor efficacy of β-lap in three subcutaneous syngeneic tumor models: MC38, TC-1, and B16 each with different NQO1 levels. In the MC38 tumor model, 25 mg/kg of β-lap was systemically (intravenously) administered to tumor-bearing WT C57BL/6 mice on day 7 after tumor inoculation.

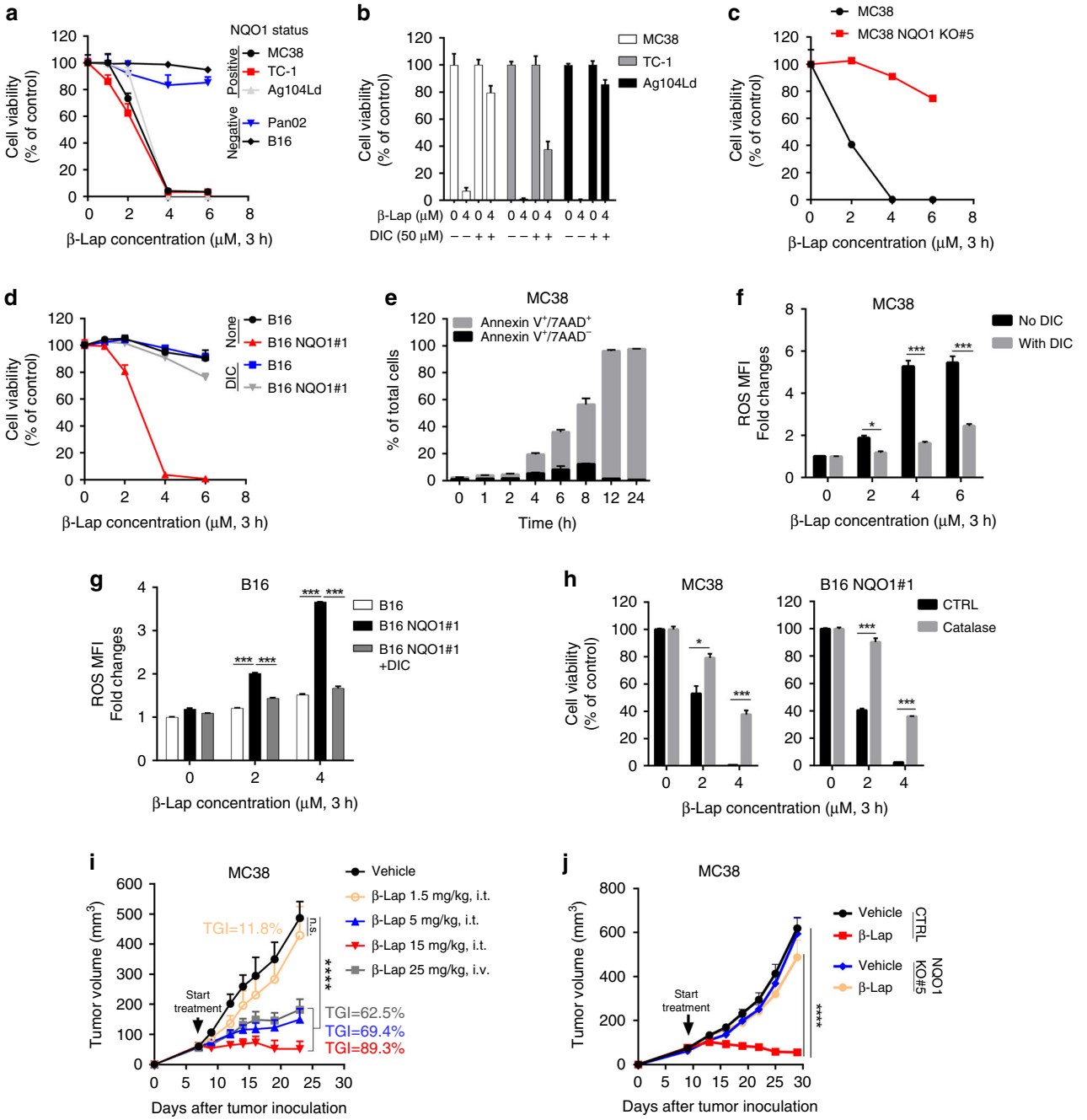

**Fig. 1** β-Lap kills murine tumor cells in an NQO1-dependent manner. **a** NQO1 positive tumor cell lines MC38, TC-1, and Ag104Ld and NQO1 negative cell lines Panc02 and B16 grown in 96-well plates were treated with β-lap (0–6 μM) for a 3 h followed by washing and replacing medium. Cell viability was determined by Sulforhodamine B (SRB) Assay 48 h later. **b** MC38, TC-1 and Ag104Ld cells were exposed to 4 μM β-lap ± dicoumarol (DIC, 50 μM) for 3 h and cell survival was assessed 48 h later. **c** MC38 cells with CRISPR-based NQO1 knockout (MC38 NQO1KO #5) planted in 96-well plates were exposed to β-lap for 3 h and cell survival was assessed 48 h later. **d** B16 cells stalely harboring a pCMV-NQO1 expression vector (B16 NQO1#1) planted in 96-well plates were exposed to β-lap ± dicoumarol (DIC, 50 μM) for 3 h purse and survival assessed 48 h later. **e** MC38 cells were exposed to a lethal dose of β-lap (4 μM) for indicated times, then stained with 7AAD and Annexin V followed by flow cytometry analysis. **f, g** MC38 cells in **f** or B16 and B16 NQO1#1 cells in **g** were exposed to β-lap ± dicoumarol (DIC, 50 μM) for 3 h and then ROS level was determined by DCFDA cellular ROS assay. **h** MC38 and B16 NQO1#1 cells were exposed to β-lap for 3 h. Catalase (1000 U/ml) was added and cell survival was assessed 48 h later. **i** C57BL/6 mice ($n = 5$/group) were transplanted with $6 \times 10^5$ MC38 cells and treated with β-lap (1.5, 5, or 15 mg/kg, intratumorally; or 25 mg/kg, i.v.) every other day for four times. **j** C57BL/6 mice were transplanted with $6 \times 10^5$ MC38 cells (NQO1 WT or KO, $n = 5$/group) and treated with β-lap (15 mg/kg, i.t.) every other day for four times. Data are shown as mean ± SEM from three independent experiments. $**p < 0.01$, $***p < 0.001$, $****p < 0.0001$ determined by unpaired Student's $t$-test in **f**– **h** or two-way ANOVA in **i** and **j**

Treatment with β-lap resulted in marked tumor inhibition (Fig. 1i). β-Lap might act on various cells when delivered systemically. To mechanismically explore whether and how β-lap functions within the local tumor environment for tumor regression, various doses of β-lap were intratumorally injected every other day for total four doses. Local treatment also significantly suppressed tumor growth in a dose-dependent manner. Tumor growth inhibition rates (TGI, %) were 62.5% for mice with systemic β-lap treatment and 11.8%, 69.4%, and 89.3% for mice with local β-lap (1.5 mg/kg, 5 mg/kg, and 15 mg/kg) treatments (Fig. 1i). Notably, local treatment with much lower dose of β-lap (15 mg/kg) induced more robust tumor regression than systemic treatment and 46.7% (7/15) of mice achieved complete tumor rejection (Fig. 1i), suggesting that major action site might be related to local tumor microenvironment. Moreover, β-lap treatment dramatically increased the ratio of Annexin $V^+/7AAD^+$ and Annexin $V^+/7AAD^-$ (Supplementary Fig. 1h) and decreased the Ki-67$^+$ populations (Supplementary Fig. 1i) in CD45-negative cells, indicating the cytotoxicity of β-lap on tumor cells in vivo. Consistent with the in vitro study, β-lap's therapeutic effect was abolished in NQO1 knockout MC38 mouse models. (Fig. 1j). Similarly, in TC-1 tumor model, β-lap treatment also led to significant tumor suppression (Supplementary Fig. 1j). To further confirm the specificity of β-lap in vivo, we established subcutaneous xenografts using parental NQO1-deficient or NQO1-overexpressing B16 cells clone (#1 and #4) in WT C57BL/6 WT mice. As expected, NQO1-null tumors did not respond to β-lap treatment (Supplementary Fig. 1k). In sharp contrast, NQO1 overexpressing (clone #1 and #4) tumor-bearing mice showed a dramatic tumor suppression after β-lap treatment (Supplementary Fig. 1k). Of note, B16 tumors overexpressing NQO1 grew much faster than B16 parent cells, indicating that NQO1 promotes in vivo tumor growth[16]. Together, these results provide evidence that β-lap selectively suppresses NQO1 positive murine tumor growth in vitro and in vivo; NQO1 is essential and sufficient for β-lap-mediated antitumor effect.

**β-Lap-mediated-antitumor effect depends on CD8$^+$ T cells**. Most studies on how β-lap kills tumor cells have focused on cancer cell-autonomous mechanism[19,20,23]. Here, we asked whether β-lap-mediated antitumor effect involves the immune system. We established MC38 tumors in immunocompetent and immunodeficient mice respectively to study the effect of β-lap on adaptive immunity. After only four doses of β-lap treatment, MC38 tumor was eradicated in WT C57BL/6 mice with 50% of mice cured (Fig. 2a). Unexpectedly, β-lap lost therapeutic activity in immune-deficient Rag1 KO (*Rag1$^{-/-}$*) C57BL/6 mice (Fig. 2a). A similar effect was observed in NQO1 overexpressing B16 tumor model (Supplementary Fig. 2a), suggesting that the adaptive immune system is required for the profound antitumor effect of β-lap in vivo. Indeed, we found an increase in CD4$^+$ and CD8$^+$ T cells in β-lap-treated tumors compared to control MC38 tumors (Supplementary Fig. 2b), suggesting that some subsets of T cells might contribute to adaptive immunity. While treatment with β-lap alone or β-lap combined with CD4$^+$ T cell depletion controlled MC38 tumor growth, CD8$^+$ T cells depletion abolished β-lap's antitumor effect (Fig. 2b, Supplementary Fig. 2c), suggesting that CD8$^+$ T cells, but not CD4$^+$ T cells, are required for β-lap-mediated tumor regression. Similar results were obtained in mice bearing TC-1 tumors (Fig. 2c). To determine whether β-lap-mediated antitumor responses result in prolonged protective T cell immunity, we rechallenged the mice that underwent complete MC38 tumor rejection after β-lap treatment with a lethal dose of MC38 cells. All the β-lap cured mice rejected the rechallenged tumors (Fig. 2d), indicating the generation of memory T cells after β-lap treatment

To test the efficacy of β-lap in controlling human tumors, we developed two xenograft models in immune-reconstituted C57BL/6 *Rag1$^{-/-}$* (Fig. 2e) and NSG-SGM3 mice (Fig. 2f). Human lung carcinoma cell line A549 that highly expresses NQO1 was sensitive to β-lap treatment in vitro (Supplementary Fig. 2d). In C57BL/6 *Rag1$^{-/-}$* immune-reconstituted model[6,24], A549 cells were s.c. inoculated. After the tumor was well established (about 100 mm$^3$), a total of $2 \times 10^6$ lymph node (LN) cells from OTI transgenic mice were transferred into tumor-bearing *Rag1$^{-/-}$* mice to reconstitute a small number of T cells without reducing tumor growth. LN cells from OTI transgenic mice contain about 98% OVA-specific T cell that cannot respond to human tumor antigens but suppress homeostatic proliferation of a small number of non-OTI T cells. A small fraction of non-OT-1 T cells have the potential to recognize human tumor antigens, which approximates 200–1000 clones, the number of tumor-reactive T cell in human patients. Without T cell transfer, β-lap treatment only partially inhibited A549 tumor growth. However, in the presence of T cells, β-lap induced more robust tumor regression with 50% of tumors completely rejected (Fig. 2e), indicating that β-lap-mediated antitumor effects are largely dependent on T cells. To study the T cell function in β-lap-mediated tumor suppression in a human-specific system, we further developed a next-generation humanized xenograft model with NSG-SGM3 mice[25]. NSG-SGM3 mice injected with human CD34$^+$ hematopoietic stem cells (Hu-NSG-SGM3) showed a robust engraftment efficiencies measured by flow cytometry using human CD45, CD3, CD4, and CD8 leukocyte markers (Supplementary Fig. 3a). Naïve NSG-SGM3 and Hu-NSG-SGM3 mice were separately inoculated with A549 cells and were treated with β-lap. In line with a murine syngeneic model, β-lap had a limited antitumor effect in A549-bearing immune-deficient NSG-SMG3 mice (Fig. 2f). Notably, in the presence of the human immune system, β-lap treatment induced profound tumor regression, indicating that the reconstituted immune system restored the antitumor effect of β-lap (Fig. 2f). We further investigated tumor-infiltrated immune cells in the tumor microenvironment, and found a significant increase of CD45$^+$ cells, and CD8$^+$ T cells but not CD4$^+$ T cells in the tumor tissues after β-lap treatment (Supplementary Fig. 3b). Together, these data reveal the necessity of T cells in β-lap-induced tumor control.

**β-Lap induces dendritic cell-mediated T cell cross-priming**. Given that CD8$^+$ T cells are essential for the antitumor efficacy of β-lap, we hypothesized that β-lap treatment increases the antigen-specific T cell response. To rule out of direct effect of β-lap on CD8$^+$ T cells, we evaluated the expression of NQO1 on CD8$^+$ T cells and the effect of β-lap on CD8$^+$ T cell survival, proliferation, and function. The results showed that neither naive CD8$^+$ T from spleen nor tumor-infiltrating CD8$^+$ T cell expressed NQO1 (Supplementary Fig. 4a). Moreover, tumor lethal dose of β-lap had no effect on CD8$^+$ T cell apoptosis and on anti-CD3/anti-CD28-stimulated cell proliferation and IFNγ production (Supplementary Fig. 4b–d). To test if β-lap increases the antigen-specific T cell response, splenocytes from MC38 tumor-bearing mice were collected 10 days after the initial β-lap treatment and an IFNγ ELISPOT assay was performed to measure the effector function of activated T cells. As shown, in the presence of tumor antigen (irradiated tumor cells), IFNγ-producing T cells dramatically increased in the β-lap treatment group (Fig. 3a). We further generated an MC38-OVA cell line using OTI peptide to better track T cell responses. Similarly, in the IFNγ ELISPOT assay with OTI peptide, the numbers of OTI-specific T cells were much higher in spleens of mice after β-lap treatment (Fig. 3b). The increase of tumor-specific CD8 cells

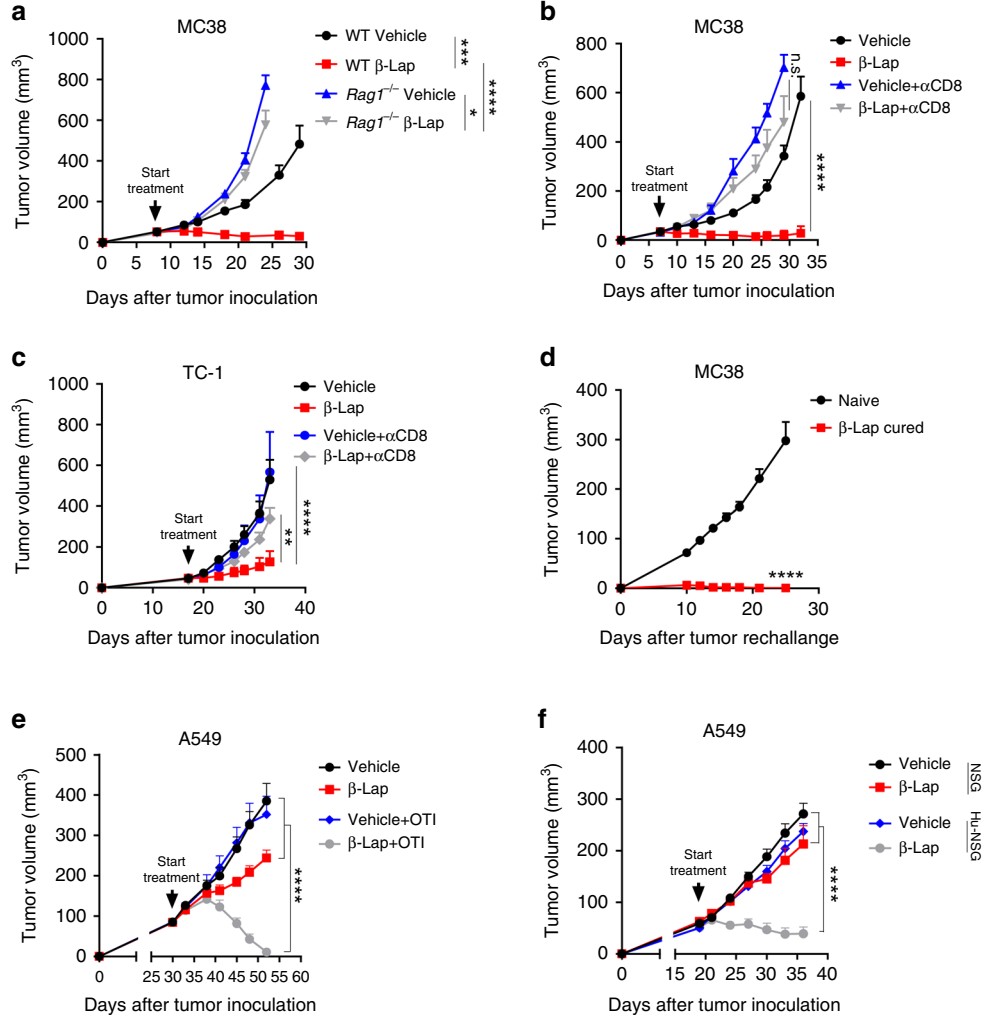

**Fig. 2** β-Lap-mediated-antitumor effect depends on CD8+ T cells. **a** $6 \times 10^5$ MC38 cells were subcutaneously transplanted into C57BL/6 WT ($n = 5$ for vehicle group; $n = 6$ for β-lap treatment group) and $Rag1^{-/-}$ mice ($n = 5$/group), respectively. Tumor-bearing mice were treated with β-lap (15 mg/kg, i.t.) every other day for four times. Numbers of tumor-free mice after treatment were shown. **b** C57BL/6 mice ($n = 5$/group) were transplanted with $6 \times 10^5$ MC38 tumor and treated with β-lap (15 mg/kg, i.t.) every other day for four times. For CD8+ T cell depletion, 200 μg of anti-CD8 antibodies were intraperitoneally injected four times at three days interval during the treatment. **c** C57BL/6 mice were transplanted with $1.5 \times 10^5$ TC-1 tumor cells and were treated with β-lap (5 mg/kg, i.t.) for four times with or without anti-CD8 antibodies ($n = 5$ for β-lap treatment group; $n = 4$ for vehicle, vehicle + αCD8, and β-lap + αCD8 groups). **d** Naïve ($n = 5$/group) and β-lap cured MC38 tumor-free ($n = 7$/group) C57BL/6 mice were rechallenged subcutaneously with $3 \times 10^6$ MC38 cells on the opposite site from the primary tumor 30 days after complete rejection, and tumor growth curve was monitored. **e** $2 \times 10^6$ A549 cells were subcutaneously injected into C57BL/6 $Rag1^{-/-}$ mice ($n = 5$ for vehicle and β-lap groups; $n = 6$ for vehicle + OTI group; $n = 7$ for β-lap + OTI group). Thirty days later, the mice were i.v. adoptively transfected with $2 \times 10^6$ lymph node cells from OT-1 transgenic mice. On the following day, tumor-bearing mice were intratumorally treated with β-lap (10 mg/kg) every other day for four times. **f** $2 \times 10^6$ A549 cells were subcutaneously injected into NSG-SGM3 ($n = 5$/group) or NSG-SGM3 harboring human CD34+ hematopoietic stem cells ($n = 5$ for Hu-NSG vehicle group; $n = 6$ for Hu-NSG β-lap group). Tumor-bearing mice were treated with β-lap (10 mg/kg, i.t.) every other day for four times. Tumor growth was measured twice a week. Data are shown as mean ± SEM from three independent experiments. **p < 0.01, ****p < 0.0001 determined by unpaired Student's t-test in **a** and **b**, or two-way ANOVA in **c–f**

suggests that β-lap treatment might induce cross-priming and reactivate T cells to control the tumor growth. Cross-presentation by APCs such as dendritic cells (DCs) or macrophages is considered the major priming mechanism to activate tumor-specific T cells. To further nail down which APCs are essential for β-lap-induced antitumor effect, we first used anti-CSF1R Ab to deplete macrophages in the tumor tissue[26]. As shown, macrophage depletion did not affect the response of MC38-bearing mice to β-lap treatment (Fig. 3c). Batf3-dependent DCs (CD8α+ or CD103+ DCs) specialize in cross-presentation of necrotic tumor cell-derived epitopes to directly activate CD8+ T cells[27–29]. $Batf3^{-/-}$ mice lack functional CD8α+ or CD103+ DCs and have impaired cell cross-

presentation activity. In WT mice, β-lap treatment induced a robust tumor regression (Fig. 3c). In stark contrast, no therapeutic effect was seen in identically treated $Batf3^{-/-}$ mice (Fig. 3d). To further address the possibility that β-lap could increase cross-presentation, we used an antigen-specific system to track the priming and activation of tumor antigen-specific T cells. WT mice bearing MC38-OVA tumors were treated with β-lap, then DCs were isolated from the tumor-drain lymph nodes (TdLN) and co-cultured with CD8+ T cells from OTI transgenic mice. IFNγ secretion was measured to evaluate the capability of DCs to prime the antigen-specific T cells. Indeed, after β-lap treatment, DCs induced more IFNγ production by OTI T cells (Fig. 3e). Together, these results suggest that

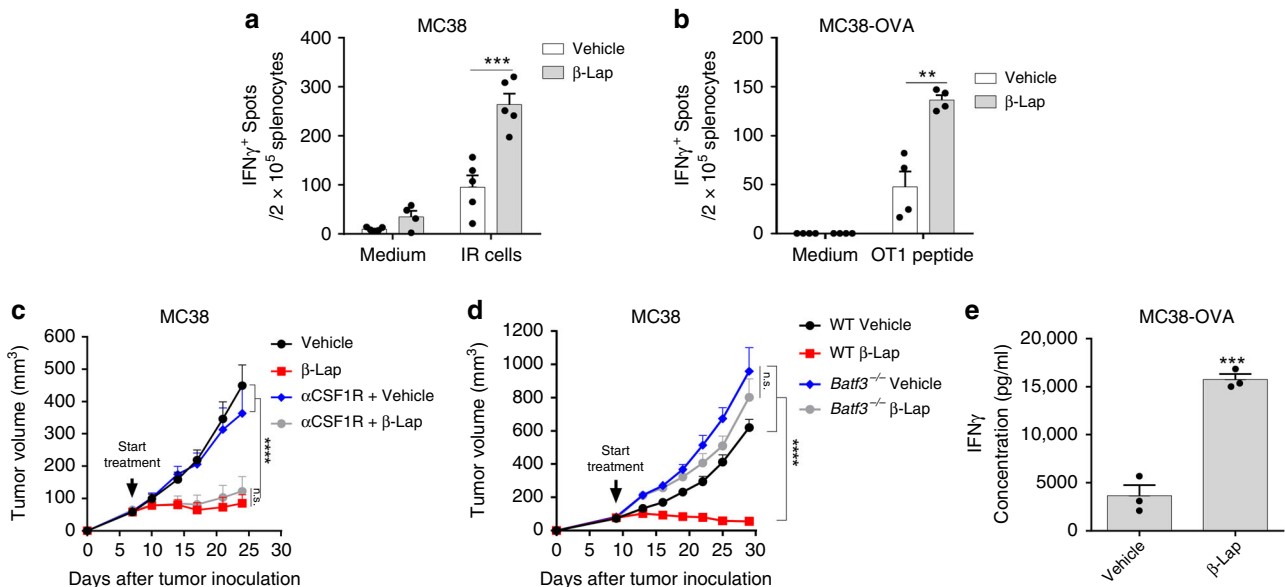

**Fig. 3** β-Lap induces Batf3-dependent dendritic cell-mediated T cell cross-priming. **a** C57BL/6 mice ($n = 5$/group) were transplanted with $6 \times 10^5$ MC38 cells and treated with β-lap (15 mg/kg, i.t.) every other day for three times, and 10 days after the first treatment, lymphocytes from the spleens were isolated and stimulated with medium or MC38 cells irradiated with 60 Gy. **b** C57BL/6 mice ($n = 4$/group) were transplanted with $1 \times 10^6$ MC38-OVA cells and treated with β-lap (15 mg/kg, i.t.) every other day for three times, and 10 days after the first treatment, lymphocytes from the spleens were isolated and stimulated with 2.5 μg/ml of OTI peptide. IFNγ-producing cells were determined by ELISPOT assay. **c** C57BL/6 mice ($n = 5$/group) were transplanted with $6 \times 10^5$ MC38 cells and treated with β-lap (15 mg/kg, i.t.) every other day for four times. One hundred micrograms of anti-CSF1R Ab were intratumorally injected three times at three days interval during the treatment. **d** $6 \times 10^5$ MC38 cells were subcutaneously transplanted into C57BL/6 WT ($n = 5$/group) and $Batf3^{-/-}$ mice ($n = 5$ for vehicle treatment group; $n = 6$ for β-lap treatment group), respectively. Tumor-bearing mice were treated with β-lap (15 mg/kg, i.t.) every other day for four times. Tumor growth was monitored twice a week. **e** MC38-OVA-bearing mice ($n = 3$/group) were treated with β-lap (15 mg/kg, i.t.) for one dose, and 4 days later, CD11c$^+$ dendritic cells were purified from the tumor drain lymph nodes, and co-cultured with CD8$^+$ T cells isolated from the spleen of OT-1 transgenic mice. The activity of cross-priming of T cells was determined by the level of cell-secreted IFNγ via Cytometric Bead Array (CBA) mouse IFNγ assay. Data are shown as mean ± SEM from three independent experiments. **$p < 0.01$, ***$p < 0.001$, ****$p < 0.0001$ determined by unpaired Student's $t$-test in **a**, **b** and **e** or two-way ANOVA in **c** and **d**

DC-mediated cross-priming is required for the β-lap-induced tumor regression and tumor reactive T cell response.

**β-Lap-mediated-antitumor effect depends on innate sensing.** APCs in the tumor microenvironment are dysfunctional, leading to ineffective priming and activation of T cells[6,30]. Type I interferon (IFN) is essential for optimal cross-priming of T cells[30,31]. We tried to determine whether type I IFN signaling was involved in T cell response mediated by β-lap treatment. Indeed, we observed upregulated IFNα/β and IFNs response genes CXCL10 as well as other cytokines such as IFNγ and TNFα in tumors treated with β-lap (Supplementary Fig. 5a, b). We further examined whether type I IFN signaling was required for the therapeutic effect of β-lap. Strikingly, blocking type I IFN signaling significantly diminished the therapeutic effect of β-lap (Fig. 4a). To determine whether tumor cell or host IFN signaling was essential, MC38 tumors were inoculated into WT and IFNAR1-deficient ($Ifnar^{-/-}$) C57BL/6 mice, followed by β-lap treatment. The results showed that the therapeutic effect was abrogated in mice with impaired IFNAR1 signaling (Fig. 4b), suggesting that host type I IFNs are required for β-lap-mediated antitumor effect. It has been demonstrated that MyD88 is involved in type I IFN production and antitumor immunity by some anticancer agents[13,32,33]. Co-culture of bone marrow-derived dendritic cells (BMDCs) and β-lap-treated tumor cells (MC38 and NQO1 overexpressing B16 cells) in vitro significantly increased the production of IFNβ protein (Fig. 4c, Supplementary Fig. 5c). MyD88 deficiency completely abolished the IFNβ production when co-culture β-lap-treated tumor cells with BMDCs from

$Myd88^{-/-}$ mice (Fig. 4c, Supplementary Fig. 5c). To further determine whether MyD88 signaling is required for β-lap treatment, MC38 tumors cells were s.c. implanted into WT and MyD88-deficient ($Myd88^{-/-}$) mice. Similarly, β-lap treatment increased the level of IFNβ in the tumor tissue in WT but not in $Myd88^{-/-}$ mice (Fig. 4d). Moreover, β-lap-induced tumor regression disappeared in MC38 bearing $Myd88^{-/-}$ mice with the same therapeutic schema (Fig. 4e, Supplementary Fig. 6a). Similarly, the antitumor effect of β-lap in NQO1-overexpressing B16 tumor models was also abolished in $Myd88^{-/-}$ mice (Supplementary Fig. 6b) or $Ifnar^{-/-}$ mice (Supplementary Fig. 6c). These results demonstrated that the host MyD88 was indispensable for the β-lap induced production of type I IFNs and its antitumor effect.

Because β-lap can induce robust tumor cell death, we hypothesized that this in turn resulted in the secretion of DAMPs and exposure of tumor antigens, thereby boosting antitumor immune responses. Interestingly, knockout of TLR4 but not TLR9, both of which are the major upstream receptors of MyD88 signaling to sense the DAMPs, led to the similar therapeutic resistance (Fig. 4f, Supplementary Fig. 6a). Previous studies have shown that HMGB1 (and HMGB1/DNA complexes), one of the endogenous ligands of TLR4, can function as a danger signal that stimulates DCs cross-priming in a MyD88-dependent fashion[33]. To determine whether β-lap-induced tumor regression is HMGB1-dependent, anti-HMGB1 mAb was administered to neutralize free HMGB1 along with β-lap treatment. The results showed that blockade of HMGB1 signaling diminished the antitumor effect of β-lap (Fig. 4g), indicating that β-lap may induce HMGB1 release in the tumor

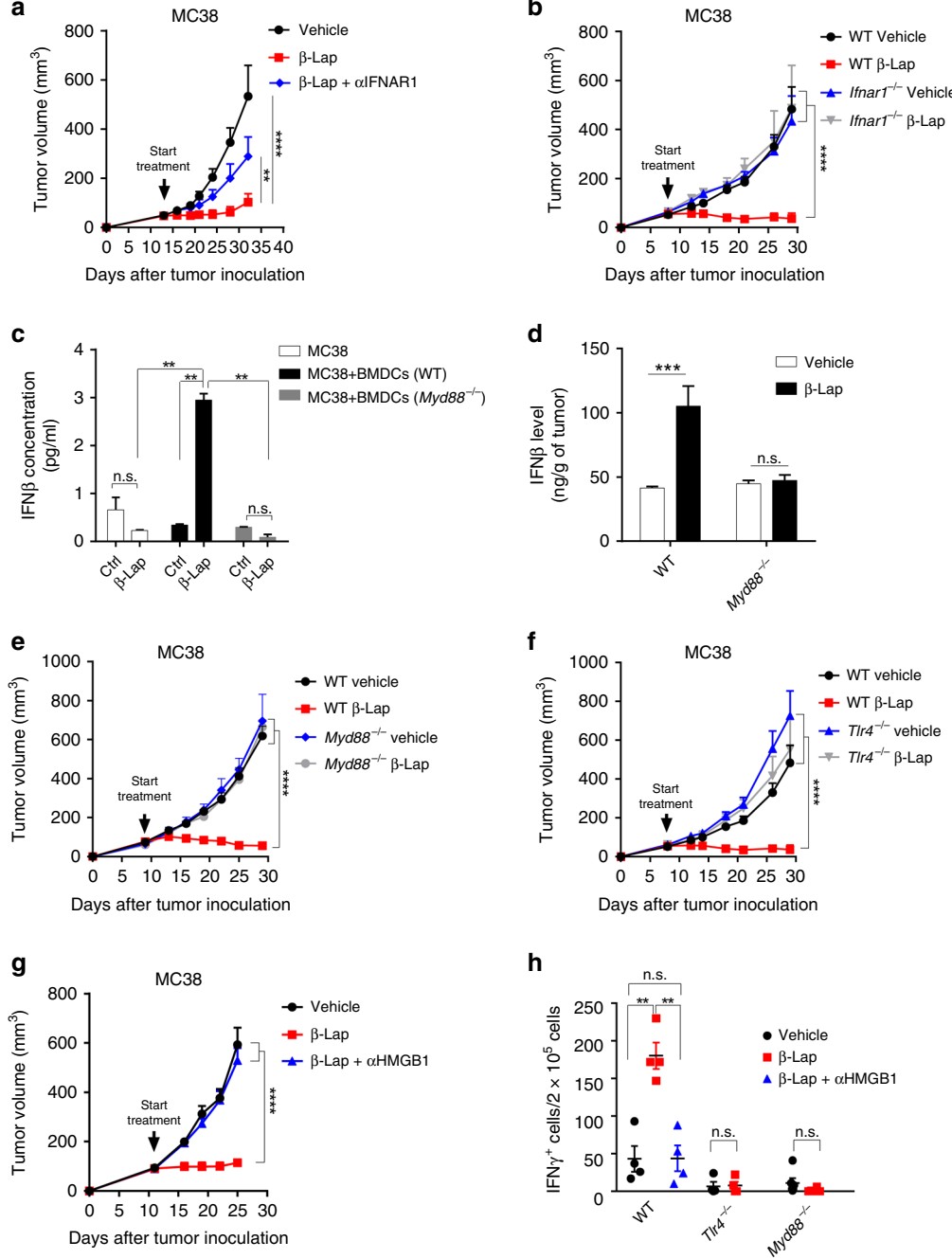

**Fig. 4** Type I IFNs and TLR4/MyD88/ signaling are required for the antitumor effect of β-lap. **a** C57BL/6 mice (n = 5/group) were transplanted with MC38 cells and treated with β-lap (15 mg/kg, i.t.) every other day for four times. Anti-IFNAR blocking antibodies (150 μg, i.t.) were administrated every four days for three times during the treatment. **b** The MC38 tumor-bearing WT (n = 5/group) and *Ifnar1[−/−]* (n = 4/group) C57BL/6 mice were treated with β-lap. **c** MC38 cells were treated with β-lap (4 μM) for 3 h followed by washing and replacing fresh medium. Twenty-four hours later, BMDCs from WT or *Myd88[−/−]* mice were co-cultured with β-lap-treated tumor cells for another 48 h. The level of IFNβ from the culture supernatant was detected by ELISA. **d** WT and *Myd88[−/−]* (n = 4/group) C57BL/6 mice were transplanted with MC38 cells and treated with β-lap. Four days later, tumor tissues were collected for protein extraction, and level of IFNβ was measured by ELISA. **e** WT (n = 5/group) and *Myd88[−/−]* (n = 3/group) C57BL/6 mice were transplanted with MC38 cells and treated with β-lap. **f** WT (n = 5/group) and *Tlr4[−/−]* (n = 4/group) C57BL/6 mice were transplanted with MC38 cells and treated with β-lap. **g** C57BL/6 mice (n = 5/group) were transplanted with MC38 cells and treated with β-lap (15 mg/kg, i.t.) every other day for four times. Anti-HMGB1 neutralized antibody (200 μg, i.p) was administrated every 3 days for three times during the treatment. **h** MC38 tumor-bearing WT (n = 4/group) or *Tlr4[−/−]* (n = 4/group) or *Myd88[−/−]* (n = 6/group) C57BL/6 mice were treated with β-lap. Anti-HMGB1 neutralized antibody (200 μg, i. p) was administrated every 3 days for three times during the treatment. Twelve days after the initial treatment, lymphocytes from the tumor drain lymph nodes were isolated and stimulated with MC38 tumor cells irradiated with 60 Gy. IFNγ-producing cells were determined by ELISPOTs assay. Data are shown as mean ± SEM from three independent experiments. *p < 0.05, **p < 0.01, ***p < 0.001, ****p < 0.0001 determined by two-way ANOVA test

microenvironment to enhance innate response via a TLR4/MyD88 pathway. To further address the essential role of HMGB1, we evaluated the tumor-specific T cell response when blocking signaling in conjunction with β-lap treatment. MC38 tumor cells were s.c. implanted into WT or $Tlr4^{-/-}$ or $Myd88^{-/-}$ C57BL/6 mice, and tumor-bearing mice were treated with β-lap with or without anti-HMGB1Ab. In WT mice, β-lap treatment increased the number of tumor-reactive T cells and this effect was abolished when co-administrated with anti-HMGB1 neutralizing Ab (Fig. 4h). Similarly, in $Tlr4^{-/-}$ and $Myd88^{-/-}$ mice there were much less tumor reactive T cells in the control group compared with that in WT mice (Fig. 4h). More importantly, β-lap treatment could not enhance the tumor-specific T cell response in these deficient mice (Fig. 4h). These results together suggest that the TLR4/MyD88/type I IFNs signaling cascade is required for β-lap induced innate and adaptive antitumor immune response.

**β-Lap induces HMGB1-dependent tumor ICD.** The HMGB1-dependent tumor-specific T cell response and HMGB1-dependent antitumor effect of β-lap in vivo suggested that β-lap could potentially induce ICD in the tumor. To test this hypothesis, we checked the ICD hallmark: HMGB1 secretion in β-lap-treated tumor cells in vitro. Indeed, we observed a dose-dependent secretion of HMGB1 in NQO1+ tumor cells but not in NQO1− cells (Fig. 5a). Inhibition of NQO1 by dicoumarol abolished β-lap-induced HMGB1 secretion. We presumed that HMGB1 exposure might dictate the immunogenicity of β-lap-induced tumor cell death. HMGB1 was found to be critical for β-lap-induced immunogenicity in three experiments: (i) BMDCs were exposed to OVA protein for 4 h, then naïve OT1 CD8+ T cells and the supernatants from β-lap-treated tumor cells (with/without αHMGB1 antibody or knockdown the HMGB1 in MC38 cells). IFNγ secretion was measured to evaluate the capability of BMDCs to prime the antigen-specific T cells. The results showed that β-lap treatment greatly increased the IFNγ secretion, and this effect was significantly diminished when HMGB1 was neutralized or knocked down (Supplementary Fig. 7a, b). (ii) Dying MC38-OVA cells induced by β-lap in vitro were injected into the flank of C57BL/6 mice in conjunction with or without anti-HMGB1Ab (Fig. 5b). The numbers of tumor antigen-specific T cells (Fig. 5c) and IFNγ production (Fig. 5d) in TdLN were determined; (iii) C57BL/6 mice that received vaccination with β-lap-induced dying MC38-OVA cells in the presence of anti-HMGB1Ab were rechallenged with MC38-OVA for evaluation of the antitumor protection (Fig. 5e, f). As shown, mice vaccinated with β-lap-induced dying cells had more IFNγ-producing antigen-specific T cells compared to mice vaccinated with living cells (Fig. 5c, d). However, the tumor-specific T cell responses diminished when vaccinated with dying cell and anti-HMGB1Ab mixture (Fig. 5c, d). Consistently, $Tlr4^{-/-}$ mice also showed reduced tumor reactive T cells and IFNγ production compared with WT mice when vaccinated with the identical dying cells (Fig. 5c, d). To test the ability of β-lap-induced dying cells to activate the adaptive immune system, we used a prophylactic tumor vaccination model in immunocompetent C57BL/6 mice (Fig. 5e). Immunization of mice with β-lap-induced dying cells prevented the growth of the rechallanged tumor (Fig. 5f). Notably, the antitumor protection effect decreased when mice were vaccinated with dying cells and anti-HMGB1 neutralized antibody (Fig. 5f). Together, these results indicate that β-lap induces ICD and enhances antitumor immunogenicity in an HMGB1-dependent manner.

**β-Lap overcomes checkpoint blockade resistance.** In clinical practice, patients with well-established tumors may generate complicated immunosuppressive networks and are generally refractory to immunotherapy[3,34]. Similarly, in our preclinical model, complete tumor rejection was achieved only in mice bearing small tumors (about 50 mm³) after β-lap treatment (TGI, 93.0%; Fig. 6a, b), and large established tumors (about 150–200 mm³) were only partially controlled by identical treatment protocols (TGI, 59.36%; Fig. 6a, b). The finding that β-lap provokes an innate and adaptive immune response as part of its mechanism of action paved a path for the combination of β-lap with T cell checkpoint blockade to eradicate the advanced and checkpoint blockade refractory tumors. To test this, we next combined local β-lap treatment with anti-PD-L1 treatment in mice bearing established large MC38 tumors. The advanced MC38 tumors only showed a moderate response to anti-PD-L1 alone (TGI, 63.25%; Fig. 6c). In stark contrast, mice in the combination groups showed robust tumor control and regression (TGI, 96.86%; Fig. 6c, Supplementary Fig. 8a). Notably, 60% of tumor-bearing mice completely rejected their tumors on combination treatment (Fig. 6c, Supplementary Fig. 8a). Interestingly, synergistic effects of β-lap and immunotherapy were also observed using a much lower dose (5 mg/kg, i.t.) of β-lap locally (Supplementary Fig. 8b). Recent studies showed that cancer immunotherapy is enhanced by local and abrogated by systemic chemotherapy treatment[35,36]. To further evaluate if systemic β-lap treatment has any immunosuppressive effects and impairs anti-PD-L1 immunotherapy, MC38 tumor-bearing mice were administrated with systemic β-lap treatment (30 mg/kg, i.p.) monotherapy or combined with anti-PD-L1 (Fig. 6d). Monotherapy of β-lap or anti-PD-L1 led to similarly moderate inhibition of tumor growth (Fig. 6e, Supplementary Fig. 8c). Combination treatment had a synergistic effect on their antitumor action with 25% of tumors completely rejected (Fig. 6e, Supplementary Fig. 8c) and markedly improved the survival of tumor-bearing mice (Fig. 6f). B16 tumors express PD-L1 but have poor immunogenicity and are not responsive to PD-L1/PD-1 immune checkpoint blockade[37,38]. We used the NQO1-overexpressing B16 tumor model to evaluate the therapeutic efficacy of β-lap and anti-PD-L1 combination treatment (Supplementary Fig. 8e). As expected, NQO1-overexpressing B16 tumors failed to respond to anti-PD-L1 Ab alone (Supplementary Fig. 8f). By contrast, β-lap monotherapy largely inhibited the growth of the B16 NQO1 tumors. Strikingly, when combined with PD-L1 blockade, β-lap had a markedly synergetic antitumor effect (Supplementary Fig. 8f).

To further elucidate the mechanism of synergy between β-lap and PD-L1 blockade, we investigated tumor-infiltrated immune cells in the tumor microenvironment by flow cytometry analysis and tracked antigen-specific T cells in the MC38-OVA tumor model. We found a significant increase of CD45+ cells, CD8+ T cells, and CD8/Treg ratio in the tumor tissues with either β-lap or anti-PD-L1 monotherapy, and these effects were dramatically magnified in the combination group (Fig. 6g). Immunostaining of CD8+ T cells further confirmed the increased tumor-infiltrating CD8+ T cells in TME (Supplementary Fig. 9). We further tracked CD8+ T cells specific for the model antigen OVA$_{257-264}$ (OTI peptide) in the tumor tissues and spleen. As shown, either β-lap or anti-PD-L1 monotherapy increased the OT1 antigen-specific T cells in the tumor tissues (Fig. 6g) and spleen (Fig. 6h). Strikingly, combined therapy robustly expanded the tumor antigen-specific T cells (Fig. 6g, h). These results suggest that β-lap treatment enhanced the tumor immunogenicity and increased T cell infiltration and tumor-specific T cell response when combined with the PD-L1 blockade. Taken together, these data demonstrate a potent synergy between β-lap and PD-L1 blockade in controlling large established and checkpoint blockade refractory NQO1-positive tumors.

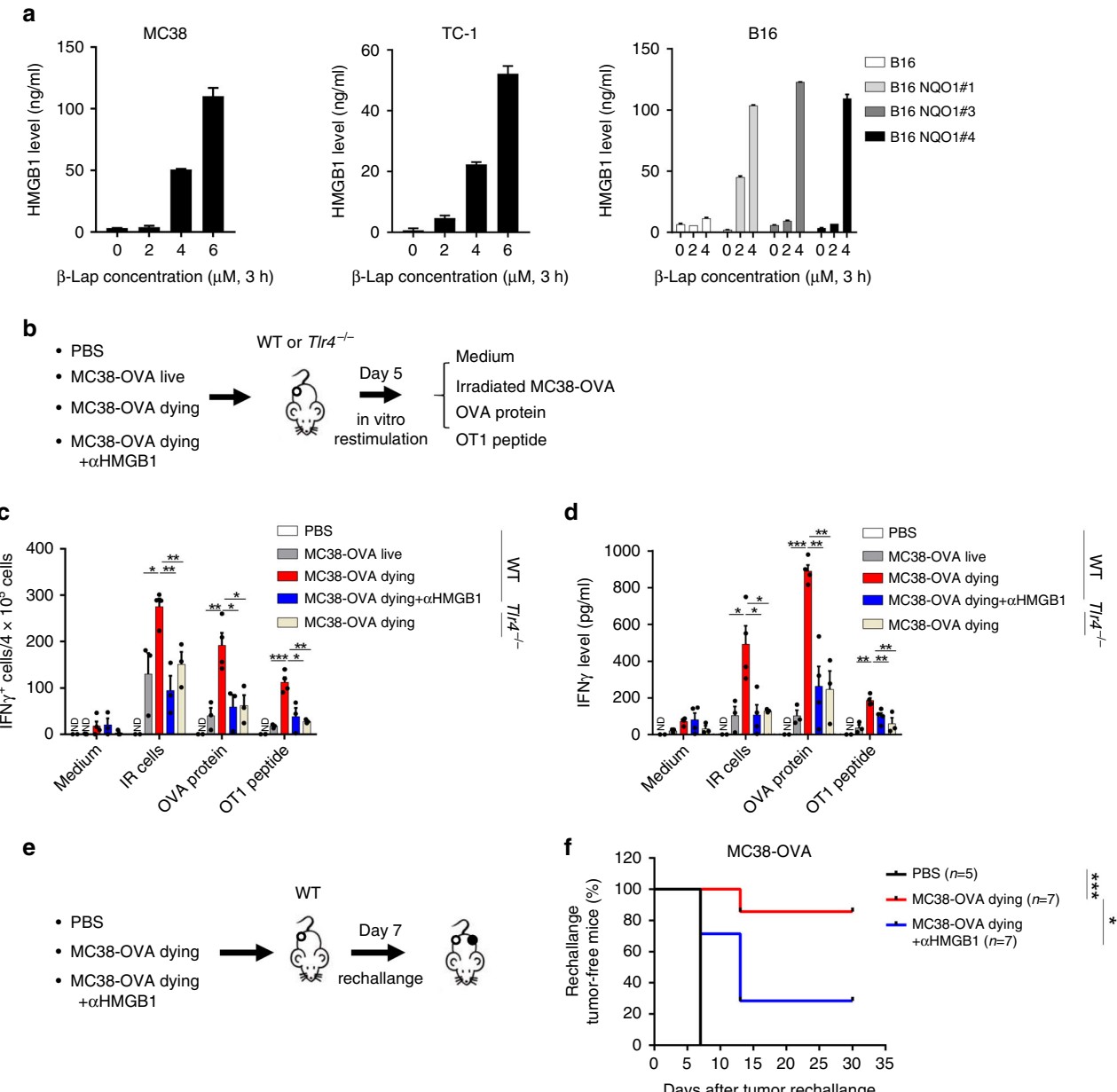

**Fig. 5** β-Lap induces HMGB1-dependent immunogenic cell death. **a** MC38, TC-1, and B16 (NQO null and overexpression clones) were treated with β-lap for 3 h followed by washing and replacing medium. The level of HMGB1 released into the culture supernatant was determined by ELISA 24 h later. **b** The research schema for in vivo cross-presentation of tumor-specific antigen from β-lap-induced dying tumor cells in panels **c** and **d**. **c**, **d** Live or β-lap-induced dying MC38-OVA cells were subcutaneously inoculated into the flank of WT or $Tlr4^{-/-}$ C57BL/6 mice ($n = 4$ for MC38-OVA dying group; $n = 3$ for other groups) along with or without anti-HMGB1 antibody. Five days later, single-cell suspensions from tumor drain lymph nodes were collected and re-stimulated with OVA protein, OT-1 peptide, or irradiated MC38-OVA cells for 48 h. IFNγ-producing cells were determined by ELISPOTs assay in **c**, and IFNγ secretion level was quantified by CBA mouse IFNγ assay in **d**. **e** The research schema for the immunogenic vaccine assay in **f**. **f** MC38-OVA cells treated with β-lap in vitro were inoculated s.c. along with or without anti-HMGB1 antibody into the flank of C57BL/6 mice ($n = 5$ for PBS group; $n = 7$ for MC38-OVA dying and MC38-OVA dying + αHMGB1 groups). After 7 days, mice were rechallenged with live MC38-OVA cells by injection into the contralateral flank. The percentage of rechallenged tumor-free mice was shown. Data are shown as mean ± SEM from three independent experiments. $*p < 0.05$, $**p < 0.01$, and $***p < 0.001$ determined by two-way ANOVA (**c**, **d**) or log-rank test in **f**

## Discussion

Lack of proper innate sensing may limit T cell-targeted immunotherapy[9,39,40]. We hypothesized that induction of immunogenic innate sensing via some tumor-targeting genotoxic agents might induce antitumor immunity and overcome immune checkpoint blockade resistance. Here we used several syngeneic immunocompetent mouse models and immune-reconstituted human xenograft models to evaluate the antitumor efficacy of

NQO1 bioactivatable β-lapachone (β-lap). We demonstrated that β-lap had an impressive antitumor effect in vivo, largely depending on innate and adaptive immunity. We discovered that after activation by NQO1, β-lap caused tumor-selective cell death and induced innate sensing for adaptive antitumor immunity. Mechanistically, tumor cells treated with β-lap triggered ICD and increased tumor immunogenicity by the release of HMGB1. This activated the innate immune response and induced a type I IFN

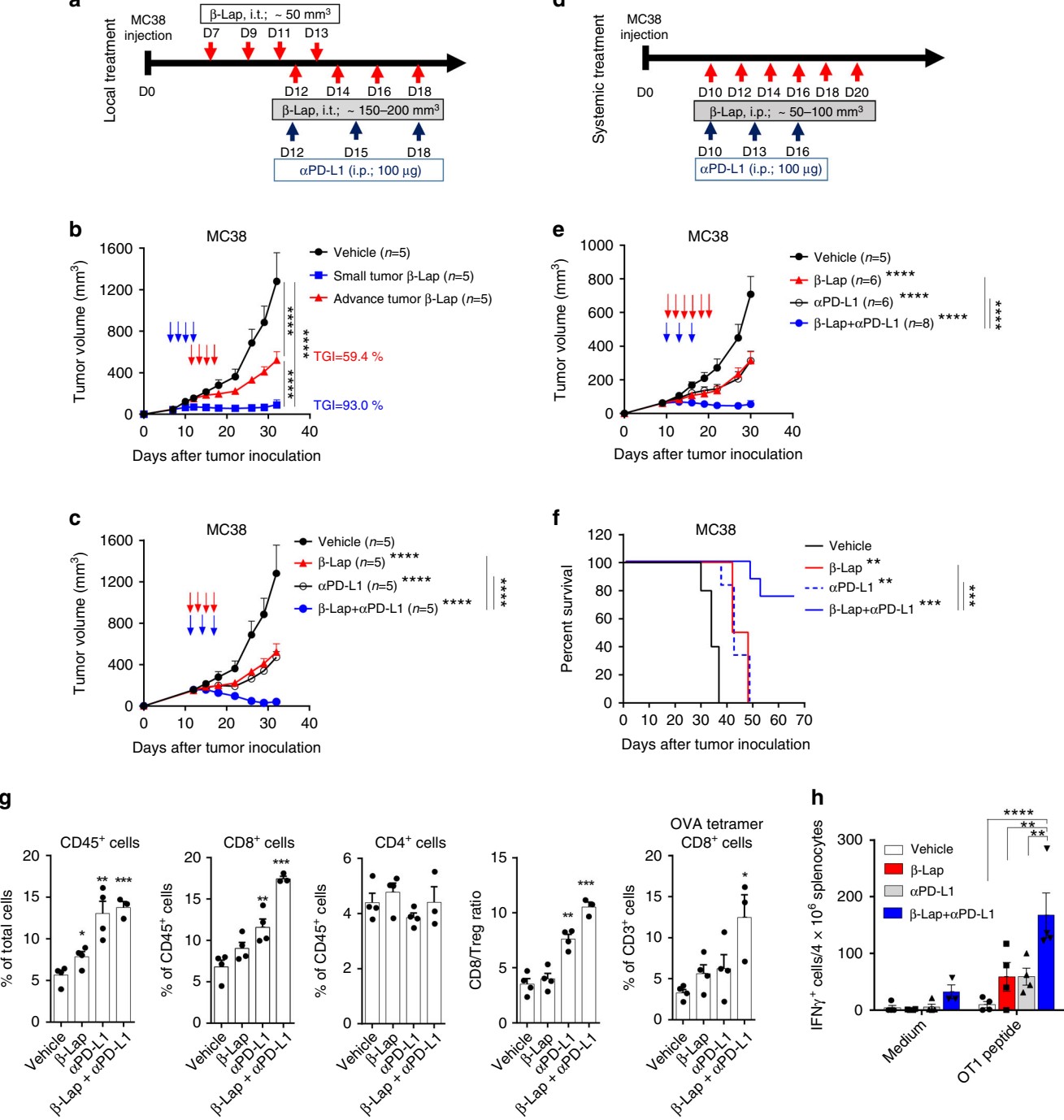

**Fig. 6** β-Lap overcomes therapeutic PD-L1/PD-1 blockade resistance. **a** Treatment schema for local β-lap treatment-based combinative therapy in panel **b** and panel **c**. **b**, **c** $6 \times 10^5$ MC38 tumor cells were s.c.inoculated into the flank of C57BL/6 mice ($n = 5$/group). Mice bearing small tumor (about 50 mm$^3$, in **b**) or advanced tumor (about 150–200 mm$^3$, in **c**) were locally treated with β-lap (15 mg/kg, i.t.) for four times with or without anti-PD-L1-based checkpoint blockage. Tumor growth was monitored twice a week, and numbers of tumor-free mice after treatment were shown. **d** Treatment schema for systemically β-lap treatment-based combinative therapy in panels **e** and **f**. **e** $6 \times 10^5$ MC38 tumor cells were s.c. inoculated into the flank of C57BL/6 mice ($n = 5$ for vehicle treatment group; $n = 6$ for β-lap and vehicle + αPD-L1 groups; $n = 8$ for β-lap + αPD-L1 treatment group). Tumor-bearing mice (about 50–100 mm$^3$) were systemically treated with β-lap (30 mg/kg, i.p.) for six times with or without anti-PD-L1-based checkpoint blockade. Tumor growth was monitored twice a week, and numbers of tumor-free mice after treatment were shown. **f** Survival curve for MC38 tumor-bearing mice with combinative treatment in panel **e**. **g**, **h** C57BL/6 mice bearing MC38-OVA tumor (about 150 mm$^3$, $n = 4$/group) were locally treated with β-lap (15 mg/kg, i.t.) every other day for four times or anti-PD-L1 (100 μg, i.p.) for three times, alone or combination. Twelve days after first treatment, tumor-infiltrating CD45$^+$ cells and lymphocytes were analyzed by flow cytometry in **g**, and OT-1 antigen-specific T cells in the spleen were determined by IFNγ ELISPOT assay in **h**. Data are shown as mean ± SEM from two to three independent experiments. *$p < 0.05$, **$p < 0.01$, ***$p < 0.001$, ****$p < 0.0001$ determined by two-way ANOVA in **b**, **c**, **e**, **g** and **h** or log-rank test in **f**

signature in TLR4/MyD88-dependent manner, which stimulated antitumor T cell adaptive immunity and restrained tumor growth. Importantly, β-lap overcame immunotherapy resistance. When combined with anti-PD-L1 mAbs, β-lap further enhanced CD8+ T cell infiltration and antigen-specific T cell response and eradicated large established and checkpoint blockade refractory tumors.

In preclinical and clinical studies, checkpoint blockade essentially takes the "brakes" off the immune system and has proven insufficient to break tolerance, unless co-administered with certain "fuels" to activate local immune activities and induce desired T cell responses[29,41,42]. Stimulating the innate immune sensing in combination with T cell checkpoint immunotherapy might be one answer. One approach to induce immunogenic innate sensing and reshape the tumor microenvironment is to use genotoxic agents such as chemotherapies that are already widely employed in cancer treatment[9,43,44]. Indeed, combining immune checkpoint blockade with chemotherapy is being extensively studied in clinical trials[12,45–47]. However, one of the major limitations in the use of traditional chemotherapy drugs arises from their lack of selectivity and related adverse toxicity to non-targeted tissues, especially the adaptive immune system. NQO1 is a two-electron oxidoreductase expressed in multiple tumor types at levels 5- to 200- fold above normal tissues, and is a potential therapeutic target[19,23]. β-Lap is a new class of NQO1-targeted prodrug which can be catalyzed and bioactivated by NQO1 to generate ROS[19,21]. Indeed, we found that β-lap selectively killed tumors highly expressing NQO1 both in vitro and in vivo, and this killing effect was abolished when we knocked out NQO1, indicating the ideal selectivity of this drug. Although genotoxic agents were assumed to exert their effects mostly via cancer cell-autonomous mechanisms for a long time, i.e., by directly inhibiting the proliferation or triggering the demise of malignant cells, accumulating evidence indicates that multiple chemotherapeutics that have been successfully employed in the clinic for decades also trigger ICD and elicit anticancer immune responses[13]. Unfortunately, most ICD-inducing agents have severe side effects at therapeutic doses due to lack of tumor-selectivity and severe immunosuppression. Importantly, β-lap distinguished itself from other chemotherapeutic agents out of its ideal target effect on NQO1-overexpressing tumors and no immunosuppression on cytotoxic immunity. We proved that tumor-infiltrated CD8+ T cell lack NQO1 and that β-lap has no cytotoxic effect on native and activated CD8+ T cell even at the tumor-lethal dose. A critical question is whether β-lap-mediated tumor-specific cell death has some interaction with the immune system; whether this "targeted" prodrug evokes an ICD and triggers innate sensing. Currently, we found that β-lap-induced NQO1+ tumor regression largely depends on host CD8+ T cells. We further proved that ß-lap induced ICD and activated innate sensing via an HMGB1/TLR4 pathway and upregulated the type I IFNs signaling in the tumor microenvironment, resulting in the promotion of Batf3 DCs to cross-prime T cells and activate of antitumoral adaptive immune response.

Reducing tumor burden and increasing tumor immunogenicity are believed to be two key factors to improve immunotherapy[48]. Notably, β-lap promoted HMGB1-depedent immunogenicity and activated innate sensing to bridge innate and adaptive immune response, and markedly shrunk the tumor mass, and is thus a promising partner for combination with immunotherapy. Encouragingly, PD-1/PD-L1 blockade has shown promising capacity to increase the proliferation and function of tumor-infiltrating CD8+ T cells, and enhance the antitumor efficacy in several cancer types[49–52]. Indeed, we demonstrated that β-lap managed to eradicate large established and checkpoint blockade refractory MC38 and B16 tumors by combination with anti-PD-

L1 immunotherapy and dramatically increased the survival rate. We further proved that combination therapy dramatically increased the tumor-infiltrating lymphocytes and tumor-antigen-specific T cells as well as the CD8/Treg ratio in the tumor microenvironment, as compared to β-lap or anti-PD-L1 monotherapy.

Overall, our study has provided an insight into how β-lap, a unique targeted prodrug, induces antitumor effect through coordinative innate and adaptive immunity and how β-lap treatment trigger innate sensing for better immunotherapy. NQO1 can be important biomarker since NQO1 is essential and sufficient for β-lap-mediated specific innate sensing and adaptive T cell responses. β-lap is currently being tested in monotherapy or in combination with the other chemodrugs in patients with NQO1+ solid tumors. Our study points out that β-lap's innate sensing capability can prepare NQO1+ patients for a successful response to immunotherapy.

## Methods

**Mice.** Female C57BL/6J and *Rag1*−/− mice were purchased from UT southwestern mice breeding core. *Myd88*−/−, *Tlr4*−/−, *Tlr9*−/−, *Batf3*−/−, and OT1 CD8+ T cell receptor (TCR)-Tg mice in the C57BL/6J background and NSG-SMG3 mice were purchased from The Jackson Laboratory. *Ifnαr1*−/− mice were provided by Dr. Anita Chong from the University of Chicago. All the mice were maintained under specific pathogen-free conditions. Animal care and experiments were carried out under institutional and National Institutes of Health protocol and guidelines. This study has been approved by the Institutional Animal Care and Use Committee of the University of Texas Southwestern Medical Center.

**Cell lines and reagents**. MC38, B16, Panc02, Ag104Ld, and A549 cells were purchased from ATCC. TC-1 cells were kindly provided by Dr. T.C. Wu at John Hopkins University. All cell lines were routinely tested using mycoplasma contamination kit (R&D) and cultured in Dulbecco's modified Eagle's medium or RPMI 1640 medium supplemented with 10% heat-inactivated fetal bovine serum, 100 U/ml penicillin, and 100 U/ml streptomycin under 5% $CO_2$ at 37 °C. β-Lapachone was synthesized as described[20] and dissolved in DMSO for in vitro study. Catalase, dicoumarol, and FTY720 were purchased from Sigma-Aldrich. OT-1 peptide and OVA protein were from ThermoFisher. Anti-CD4 (GK1.5), anti-CSF1R (AFS98), anti-IFNAR1 (MAR1-5A3), anti-PD-L1 (10F.9G2), anti-CD8 (YTS), and anti-HMGB1 mAbs were purchased from BioXCell.

**Sulforhodamine B (SRB) cytotoxicity assay**. Four thousand cells were planted in a 96- or 48-well plate with triplicates. After overnight growth, cells were exposed to β-lapachone with or without NQO1 inhibitor dicoumarol (50 μM) for a 3-h pulse. After that, the cell supernatant was replaced by fresh medium, and the plate was incubated at 37 °C in a humidified incubator with 5% $CO_2$ for another 2 or 4 days. Following treatment, the culture supernatant was removed, and fixative reagents were gently added to each well. The wells were washed with water and the plate was air dried overnight. 100 μL SRB Dye solution was added and incubated for 30 min, followed by washing and air dry. The cell growth was determined by the absorbance at 560 nM in a microplate reader (SpectroSTAR Nano, BMG labtech). % Cell growth = (100 × (cell control−experimental)) ÷ (cell control).

**NQO1 knockout and overexpression**. NQO1 gene in MC38 cells was knocked out by CRISPR/Cas9 technology. The guide sequence 5′-TTGTGTTCGGCCACAA TATC-3′ was cloned into pSpCas9 (BB)-2A-Puro plasmid (Addgene, # 62988) containing a puromycin selection gene. The plasmid was transiently transfected into MC38 cells. Forty-eight hours after transfection, puromysin-resistant cells were selected and subcloned under the selective culture medium. MC38 Cell clones (#1, #2, and #5) without NQO1 expression were used for in vitro and in vivo studies. For NQO1 overexpression, B16 cells (NQO1 null) was transiently transfected with a full-length mouse NQO1 protein expression vector pCMV3-HA-NQO1 (Sino Biological, #MG57522-CY). The NQO1 stable expressing cells were selected and subcloned under the hygromycin containing culture medium. B16 cells clones (#1, #3 and #4) with NQO1 stable expression were used for following studies. The NQO1 expression levels were determined by western blotting assay.

**Tumor growth and treatment**. Approximately $6 × 10^5$ MC38 cells or $1.5 × 10^5$ TC-1, or $1.5 × 10^5$ B16 cells were subcutaneously inoculated into the right flank of mice. Tumor-bearing mice were randomly grouped into treatment groups when tumors grew to certain sizes. For β-lap monotherapy, tumor-bearing mice were treated with β-lap locally (intratumorally, 1.5, 5, or 15 mg/kg every other day for four times) or systemically (intravenously or intraperitoneally, 25 or 30 mg/kg every other day for four or six times). For CD4 and CD8+ T cell depletion, 200 μg of antibodies were intraperitoneally injected four times at three days interval. For

mMcrophage depletion, 100 µg of Anti-CSF1R mAb were intratumorally injected three times at 3 days interval during β-lap treatment. For type I IFN blockade experiment, 150 µg of anti-IFNAR1-blocking mAbs were intratumorally injected at 3 days interval for a total three times. The blocking and depletion experiments above started one day before the first β-lap treatment. For HMGB1 blockade experiments, 200 µg of anti-HMGB1 mAbs were administered intraperitoneally (i.p.) every 3 days for total three times staring at the same day of the first β-lap treatment. For PD-L1 checkpoint blockade combination therapy, 100 µg (for the MC38 model) or 150 µg anti-PD-L1 (for the B16 model) was administered intraperitoneally to tumor-bearing mice every 3 days for total three times starting at the same day of the first β-lap treatment. Tumor volumes were measured at least twice weekly and calculated as $0.5 \times$ length $\times$ width $\times$ height. The "Spaghetti plots" of each tumor growth curve of individual mouse for in vivo study are shown in Supplementary Fig. 10.

**Immune-reconstituted mouse models**. For C57BL/6 $Rag1^{-/-}$ immune-reconstituted model, $2 \times 10^6$ A549 cells were s.c. inoculated into female $Rag1^{-/-}$ mice. After the tumor was well established (about 100 mm³), $2 \times 10^6$ total LN cells from OTI transgenic mice were intravenously injected into the tumor-bearing mice one day before treatment. Later, the mice were treated with β-lap locally (i.t., 10 mg/kg) every other day for four times. Tumor volumes were measured at least twice weekly.

For NSG-SGM3 humanized mouse model, four-week-old NSG-SGM3 female mice were irradiated with 100 cGy (X-ray irradiation with X-RAD 320 irradiator) one day prior to human CD34⁺ cells transfer. Irradiated mice were treated with Bactrim (Aurora Pharmaceutical LLC) water for 2 weeks. Cord blood was obtained from UT Southwestern Parkland Hospital. Human CD34⁺ cells were purified from cord blood by density gradient centrifugation (Ficoll® Paque Plus, GE healthcare) followed by positive immunomagnetic selection with anti-human CD34 microbeads (Stem Cell). $1 \times 10^5$ CD34⁺ cells were intravenously injected into each recipient mouse. Eleven weeks after engraftment, humanized mice with over 60 % human CD45⁺ cells reconstitution and age and sex matched non-humanized mice were inoculated with $3 \times 10^6$ A549 tumor cells subcutaneously on the right flank. At day 19, the tumor-bearing mice were treated with β-lap locally (i.t., 10 mg/kg) every other day for four times. Tumor volumes were measured at least twice weekly. All experiments were performed in compliance with UTSW Human Investigation Committee protocol and UTSW Institutional Animal Care and Use Committee.

**HMGB1 release detection**. Tumor cells were planted in a six-well plate and grown to 70% confluence and treated with increasing concentration of β-lap for 3 h, followed by washing and medium replacement. The supernatant was assayed for extracellular HMGB1 24 h later using an ELISA KIT (Chondrex).

**IFNγ enzyme-linked immunosorbent spot assay (ELISPOT)**. Tumor drain LNs and spleen from tumor-bearing mice were collected and single-cell suspension was prepared. Irradiated tumor cells or OT-1 peptides were used to re-stimulate the tumor-specific T cells. In general, a total of $2$–$4 \times 10^5$ LN cells or splenocytes and $2$–$4 \times 10^5$ irradiated tumor cells were co-cultured for 48 h, and ELISPOT assay was performed using the IFNγ ELISPOT kit (BD Bioscience) according to the manufacturer's instructions. Spots were calculated by an ImmunoSpot Analyzer (Cellular Technology Limited).

**Cell isolation from tissues**. CD11c⁺ DCs or CD8⁺ T cells were isolated from lymph nodes or spleen of mice with a positive CD11c isolation kit or a negative CD8 isolation kit (Stemcell) according to the manufacturer's instructions. For tumor single-cell suspension, tumor tissues were cut into small pieces, and resuspended in digestive buffer (1.5 mg/ml type I collagenase and 100 µg/ml DNase I) for 45 min in a 37 °C shaking incubator. After digestion, cells were passed through a 70-µm cell strainer.

**Flow cytometric analysis**. Tumor cell suspension was blocked with the anti-CD16/32 antibody (clone 2.4G2) for 10 min, and then incubated with indicated antibody for 30 min at 4 °C in the dark. Fixable viability Dye eFlour 506 (eBioscience) was used to exclude the dead cells. Sample was analyzed on a CytoFLEX (Backman coulter) flow cytometer. An example of gating schematics was shown in supplementary Fig. 11 to characterize the immune cell infiltrates in tumor tissue. Antibodies used in this study had been listed in the Supplementary Table S1.

**DCFDA cellular ROS detection assay**. The level of cellular ROS was determined by the DCFDA-Cellular ROS Detection Assay Kit (Abcam) according to the manufacturer's instructions. Briefly, cells were plated into 12-well plates and grown to about 70% confluence, and stained with DCFDA at 37 °C for 30 min. After that cells were treated with different concentration of β-lap for 3 h. ROS signal was determined using Flow cytometry at Ex/Em: 485/535 nm.

**Quantitative real-time PCR**. Total RNA from cells was extracted with the TRIzol (Invitrogen) and reversed-transcribed with iScript™ gDNA Clear cDNA Synthesis Kit (Bio-Rad). Real-time PCR was performed with SsoAdvanced™ Universal SYBR® Green Supermix (Bio-Rad) according to the manufacturer's instructions. Primers used in this study are listed in the Supplementary Table 2. β-Actin was used as the internal control. $2^{-\Delta\Delta Ct}$ method was used to calculate relative expression changes.

**Statistical analysis**. All the data analyses were performed with GraphPad Prism statistical software and shown as mean ± SEM. $*p < 0.05$, $**p < 0.01$, $***p < 0.001$ and $****p < 0.0001$ determined by two-way ANOVA or unpaired two-tailed $t$-tests. A value of $p < 0.05$ was considered statistically significant.

**Reporting Summary**. Further information on research design is available in the Nature Research Reporting Summary linked to this article.

## Data availability

The source data for the in vivo studies underlying Figs. 1–6 and Supplementary Figs. 1–8 are provided as a Source Data file. All the other data supporting the findings of this study are available within the article and its supplementary information files and from the corresponding author upon reasonable request. A reporting summary for this article is available as a Supplementary Information file.

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

## Acknowledgements

We thank Casey Timmerman for help with manuscript preparation. We thank the UT Southwestern animal breeding core facility. We thank Dr. Jinming Gao and Dr. Zhaohui Wang's help of getting the immunofluorescence imaging data. Y.-X.F. holds the Mary Nell and Ralph B. Rogers Professorship in Immunology. This study was supported in part by the NIH through National Cancer Institute grants CA141975, Texas CPRIT grant RR150072 (CPRIT scholar in Cancer Research) to Y.-X.F.

## Author contributions

X.L., Z.L., X.H., and Y-X.F. designed the experiments, analyzed the data, and wrote the manuscript. X.H. and Y-X.F. supervised the project. X.L., Z.L., A.Z., and L.J. conducted the experiments. C.H., A.S, J.Q., Y.W., and D.A.B. contributed to regents/material and provided helpful discussion.

## Additional information

**Competing interests:** The authors declare no competing interests.

