## [Transparent Peer Review File · Nature Communications]

Reviewers' comments:

Reviewer #1 (Remarks to the Author):

In the present paper, the authors study the effect of NQO1-based tumor targeting prodrug, β -lapachone on induction or triggering of innate sensing to overcome the checkpoint blockade resistance in cancer cells. Several interesting observations reported in the manuscript show that β -lapachone induces immunogenic cell death, leading to the TLR4/MyD88 signaling pathway leading to type 1 interferon signaling and inducing Batf3 dendritic cell dependent cross-priming, resulting in innate and adaptive immune responses. All the experimental procedures are clearly explained, and the results shown in figures support the authors, interpretations and conclusions. Overall the study is robust and well done. However, there are several concerns which need to be clarified and addressed.

Major comments:

1. In this manuscript, there are rather many statements like 'data not shown'. I think it would be better to delete the expression 'data not shown' or show them.
2. As described in introduction section, β -lapachone is NQO1 bioactivatable drug. In Figure 1a and h, β -lapachone caused cell death in NQO1-deficient B16 cells, but β -lapachone did not show cancer cell death in Figure 1d. The authors should clarify the data.
3. The first title in the Result is "NQO1-targeting prodrug β -lap suppresses murine tumor growth in a NQO1 dependent manner both in vitro and in vivo". It is not accurate to refer the in vitro cell death as suppression of tumor growth. The authors and other investigators have already published numerous papers showing that β -lapachone suppresses tumor growth in NQO1 dependent manner. Given that this is already a well-known fact, this reviewer felt it is not necessary to discuss this in detail consuming one and half pages of the manuscript. I recommend to make this section shorter and use this space to show "data not shown" or the "supplementary figures".
4. The authors obtained data using mainly NQO1-expressing MC38 and TC-1 cell lines. The authors should perform a loss-of-function and gain-of-function studies to clarify the data in Figures 2-6. For example, in Figure 2c and supplementary Figure 1h, the author used TC-1 cells to explain β -lapachone-induced inhibition of tumor growth in vivo by CD8⁺ T cells. The authors should show the effect of β -lapachone on the inhibition of tumor growth by CD8⁺ T cells using NQO1 deficient MC38 cells to clarify the data in Figure 2a and b.
5. In Supplementary Figure 3b and Figure 6g, the authors show infiltration of T cells using flow cytometry. The author should show immunohistochemistry data to clarify T cell infiltration of the tumors. The authors also only show graphs for tumor growth in Figures. In my opinion, immunohistochemical data showing β -lapachone-induced cytotoxicity in tumors is needed to clarify and support the graph data.
6. In Figure 4G, the authors should clarify whether β -lapachone induces HMGB-DAMP signal pathway to activate of CD8⁺ T cells using cells which is knockdown of HMGB1.

Reviewer #2 (Remarks to the Author):

NQO1-based tumor targeting prodrug triggers innate sensing to overcome checkpoint blockade resistance

The authors have done a heroic amount of work in this manuscript

I have some specific questions that need addressing

In general, I would like to see the cell number used in each in vivo tumor model (s.c) displayed at the top of each figure. Some of the tumor growth curves are different from figure to figure using the same cell lines and this is sometimes due to injected cell number and would help the reader greatly.

Also, "Spaghetti plots" of each tumor grown in individual mice should be shown in the supplemental data so growth of each tumor can be visualized. Error bars are impressively small given the low number of mice used in each group as these lines are known to have sizable variation in B6 mice. Spaghetti plots would help here.

Fig. 1C - Are these MC38 "null" or "knock-down" for NQO1? Western analysis should be complemented with NGS analysis since there is still b-Lap dependent death of MC38 cells

Fig 2A,B - Why do MC38 grow so differently in WT Vehicle mice in Fig. 2A v 2B? i.e 300mm in A at d25 versus >200mm in B? I ask this because the error in B is very small, yet if you look at A there would be 50% error.....

The MC38 growth in the b-Lap treated group between Fig 6C and 4C look very different. 4C looks like a cure whereas 6C requires anti-PD1L for a cure. Why is this?

Fig 6 E and F legends are swapped around. E is the survival plot.

What is the effect of b-Lap on metastasis? spontaneous metastasis such as E0771-LMB, or 4T1?

Response to reviewer:

Reviewer #1: In the present paper, the authors study the effect of NQO1-based tumor targeting prodrug, β -lapachone on induction or triggering of innate sensing to overcome the checkpoint blockade resistance in cancer cells. Several interesting observations reported in the manuscript show that β -lapachone induces immunogenic cell death, leading to the TLR4/MyD88 signaling pathway leading to type 1 interferon signaling and inducing Batf3 dendritic cell dependent cross-priming, resulting in innate and adaptive immune responses. All the experimental procedures are clearly explained, and the results shown in figures support the authors, interpretations and conclusions. Overall the study is robust and well done. However, there are several concerns which need to be clarified and addressed.

Major comments:

Comment: 1. In this manuscript, there are rather many statements like 'data not shown'. I think it would be better to delete the expression 'data not shown' or show them.

Response: We appreciated the reviewer's suggestion. In the revised manuscript, we deleted the expression 'data not shown' and provided more details of the data in the supplementary figures.

Comment: 2. As described in introduction section, β -lapachone is NQO1 bioactivatable drug. In Figure 1a and h, β -lapachone caused cell death in NQO1-deficient B16 cells, but β -lapachone did not show cancer cell death in Figure 1d. The authors should clarify the data.

Response: We agree with the reviewer that we should clarify the data that were generated from different assays and with different treatment times. We examined several cancer cell lines with or without NQO1 expression to confirm the tumor targeting effect of β -lap. We also used cell lines that were reconstituted or knocked out of NQO1 expression to confirm the targeting effect. In Fig. 1a, 4000 cells (per well) in 48-well plates were treated with β -lap (0-8 μ M) for a 3-hr followed by washing and replacing medium. 4 days later, cell survival was determined by SRB cytotoxicity assay. The results showed that tumor cell lines (MC38, TC-1 and Ag104Ld) that express high level of NQO1 were sensitive to β -lap exposure with a lethal dose around 2-4 μ M. In contrast, NQO1 deficient cell lines, B16 and Panc02 were resistant to β -lap exposure. Only much higher concentration of β -lap ($\geq 8 \mu$ M) had obvious cytotoxicity which might result from

“off target” effect. In Fig. 1d, 4000 cells (per well) in 96-well plates were exposed to β -lap for 3 hr and cell survival was assessed 48 hr later with SRB cytotoxicity assay. Here, we did not use 8 μ M of β -lap treatment because of the potential off target effect in this experiment. In both figure 1a and 1d, 4 μ M of β -lap treatment almost killed all the NQO1 positive cells but had no effect on the negative cell lines. The variation for 6 μ M treatment might be because of the different treatment time and procedure (48 well vs 96 well, 4 days vs 2 days). In our revised manuscript, we deleted the 8 μ M treatment group in figure 1a.

Comment: 3. The first title in the Result is “NQO1-targeting prodrug β -lap suppresses murine tumor growth in a NQO1 dependent manner both in vitro and in vivo”.

It is not accurate to refer the in vitro cell death as suppression of tumor growth.

The authors and other investigators have already published numerous papers showing that β -lapachone suppresses tumor growth in NQO1 dependent manner. Given that this is already a well-known fact, this reviewer felt it is not necessary to discuss this in detail consuming one and half pages of the manuscript. I recommend to make this section shorter and use this space to show “data not shown” or the “supplementary figures”.

Response: Thanks for your suggestion. We agree with the reviewer that the first title should be revised. In page 5, lines 2-3, the first title was changed to “NQO1-targeting prodrug β -lap induces murine tumor cell death in vitro and suppresses murine tumor growth in mice in a NQO1 dependent manner”.

We agree with the reviewer that numerous papers have shown that β -lap suppresses tumor growth in NQO1 dependent manner. However, all the previous studies, including our own studies, mainly focused on the tumor-direct killing effect in vitro and in vivo by using human tumor lines and human xenograft immunodeficient model, with no mechanisms and evidence related to the host immune responses. Our current study is actually the first time to report the antitumor efficacy of β -lap *in vivo* through NQO1-dependent immune-mediated killing in immune competent hosts. Syngeneic murine tumor models allow us to better understand the role of host immunity during β -lap treatment. Therefore, we need to screen and test various murine lines that allow us to do gain-of-function or loss-of-function studies in immune competent hosts. We would like to clarify the NQO1-dependent killing effect in the murine tumor lines in vitro

first, then we are able to use NQO-1 positive lines to study the immunologic mechanisms of β -lap in tumor killing with immune competent hosts.

Comment: 4. The authors obtained data using mainly NQO1-expressing MC38 and TC-1 cell lines. The authors should perform a loss-of-function and gain-of-function studies to clarify the data in Figures 2-6. For example, in Figure 2c and supplementary Figure 1h, the author used TC-1 cells to explain β -lapachone-induced inhibition of tumor growth in vivo by CD8⁺ T cells. The authors should show the effect of β -lapachone on the inhibition of tumor growth by CD8⁺ T cells using NQO1 deficient MC38 cells to clarify the data in Figure 2a and b.

Response: Thanks for your comment. In this study, we demonstrated that β -Lap mediated-antitumor effect depended on NQO1 and CD8⁺ T cells by using MC38 (NQO1 high), TC-1 (NQO1 high), B16 (NQO1 null), MC38 NQO1 KO and B16 NQO1 overexpressing cell lines . In MC38 models, β -lap treatment induced robust tumor regression in immunocompetent mice (**Fig.1j**), while the therapeutic effect was abolished in both NQO1 KO MC38 (loss-of-function) tumor bearing mice (**Fig. 1j and fig.2a**) and MC38 tumor bearing immunodeficient mice (**Fig.2a,b**). Considering the nonresponse of NQO1 knockout MC38 tumor in immunocompetent mice, we did not further deplete the CD8⁺ T cells in WT mice and test the antitumor effect of β -lap. Instead, we used B16 model to do gain of function study. B16 tumor was not response to β -lap treatment in WT mice (**Supplementary fig.1i**), while we observed that NQO1 overexpressing B16 tumors showed a dramatic tumor regression in WT mice after β -lap treatment (**Supplementary Fig.1i and Supplementary fig.2a**). However, this therapeutic effect was diminished in immunodeficient *Rag1*^{-/-} mice (**Supplementary Fig.2b; supporting figure R1 a,b**). In line with the finding in MC38 models (**Fig.4e**), we further observed that the antitumor effect of β -lap was impaired in B16-NQO1 bearing *Myd88*^{-/-} mice (**supporting figure R1 c**). Taken together, we have demonstrated the essential role of NQO1 in β -lap triggered immune-mediated tumor killing with loss-of-function and gain-of-function studies.

The figure (Fig.R1c) has been added to the **Supplementary Fig. 6b**. In pages 9, lines 9-11 “Moreover, β -lap induced tumor regression disappeared in both MC38 and NQO1 overexpressing B16 tumor-bearing *Myd88*^{-/-} mice with the same therapeutic schema (Fig. 4e, Supplementary Fig. 6a,b)” will be added to describe this.

Supporting figure R1: β -lap suppresses NQO1-overexpressing B16 tumor growth in host immune system and Myd88 signaling dependent fashion. (a,b) NQO1-overexpressing B16 (#1) tumor bearing C57BL/6 WT (a, n=5/group) and *Rag1*^{-/-} mice (b, n=5/group) were treated with β -lap (15 mg/kg, i.t.) every other day for four times. (c) NQO1-overexpressing B16 (#1) tumor bearing *Myd88*^{-/-} (n=5/group) C57BL/6 mice were treated with β -lap (15 mg/kg, i.t.) every other day for four times. Tumor growth was monitored twice a week.

Comment: 5. In Supplementary Figure 3b and Figure 6g, the authors show infiltration of T cells using flow cytometry. The author should show immunohistochemistry data to clarify T cell infiltration of the tumors. The authors also only show graphs for tumor growth in Figures. In my opinion, immunohistochemical data showing β -lapachone-induced cytotoxicity in tumors is needed to clarify and support the graph data.

Response: We appreciate the reviewer's suggestion. As suggested, we detected the CD8⁺ T cell infiltration by immunofluorescence staining. Similarly, we observed that only few CD8⁺ T cell had infiltrated primary tumor in vehicle treatment group, β -lap or anti-PD-L1 monotherapy induced more CD8⁺ T cells infiltration, and combination treatment dramatically magnified the infiltration of CD8⁺ T cells (**supporting figure R2**).

The figure (Fig.R2) has been added to the Supplementary Fig. 9. In pages 11, lines 32-33 "Immunostaining of CD8⁺ T cells further confirmed the increased tumor infiltrating CD8⁺ T cells in TME (Supplementary Fig. 9)." will be added to describe this.

Supporting figure R2: β -Lap monotherapy or combined with anti-PD-L1 treatment increases the $CD8^+$ T cell infiltration. C57BL/6 mice bearing MC38-OVA tumor were locally treated with β -lap (15 mg/kg, i.t.) every other day for four times or anti-PD-L1 (100 μ g, i.p.) for three times, alone or combination. 12 days after first treatment, tumor infiltrating $CD8^+$ T cells were analyzed by Immunofluorescence. Scale bar is 10 μ m.

We agree with the reviewer that we should determine the cytotoxicity of β -lap to tumors *in vivo* to support the tumor growth data. According to this comment, we have further detected the Annexin $V^+/7AAD^+$ and Annexin $V^+/7AAD^-$ as well as Ki-67 $^+$ cells in the tumor tissues 18 hours after β -lap treatment. We quantified the CD45 negative cells which were mainly tumor cells. The results showed that β -lap treatment dramatically increased the ratio of Annexin $V^+/7AAD^+$ and Annexin $V^+/7AAD^-$ in CD45 negative cells (**supporting figure R3a**). Meanwhile, the Ki-67 positive cells, indicating the proliferated tumor cells, were significantly reduced (**supporting figure R3b**) after β -lap treatment. These data indicated the β -lap-induced cytotoxicity in tumor cells *in vivo*.

The figure (Fig.R3a-c) has been added to the Supplementary Fig. 1h,i. In pages 5-6, lines 1-3 “Moreover, β -lap treatment dramatically increased the ratio of Annexin $V^+/7AAD^+$ and Annexin $V^+/7AAD^-$ (Supplementary Fig. 1h) and decreased the Ki-67 $^+$ populations (Supplementary Fig. 1i) in CD45 negative cells, indicating the cytotoxicity of β -lap on tumor cells *in vivo*.” will be added to describe this.

Supporting figure R3: β-Lap induces cytotoxicity in tumor cells in vivo. C57BL/6 mice bearing MC38 tumor (n=6-8/group) were locally treated with β-lap (15 mg/kg, i.t.). 18 hours after treatment, tumor tissues were processed into single cell suspension and CD45⁺ cells were stained with 7AAD/Annexin V (a) or Ki-67 (b, c) followed by flow cytometry analysis.

Comment: 6. In Figure 4G, the authors should clarify whether β-lapachone induces HMGB-DAMP signal pathway to activate of CD8⁺ T cells using cells which is knockdown of HMGB1.

Response: We appreciate the reviewer's suggestion. As suggested, we used lentivirus-mediated shRNA to knock down the HMGB1 expression in MC38 cells (**Supporting figure R4a**). To determine whether the HMGB1-DAMP signaling pathway was required for inducing the maturation of dendritic cells and further enhancing the cross-priming of T cells after β-lap treatment, we performed the tumor-BMDCs co-culture assay. BMDCs from C57BL/6 WT mice were co-cultured with β-lap treated MC38 cells with or without HMGB1 neutralized antibody exposure. The results showed that BMDCs expressed significantly higher level of maturation marker CD40 and CD86 when co-cultured with β-lap treated MC38 cells (**Supporting figure R4b**). HMGB1 neutralization or knockdown of HMGB1 in MC38 cells dramatically impaired the CD40 and CD86 upregulation (**Supporting figure R4b**), indicating that the β-lap-induced HMGB-DAMP signal was essential for activating the innate sensing and the maturation of dendritic cells. To further address the role of HMGB1-DAMPs signaling in β-lap-induced cross-priming of CD8⁺ T cells, BMDCs were exposed to OVA protein for 4 hours, then naïve OT1

CD8⁺ T cells and the supernatants from β -lap-treated tumor cells (with/without α HMGB1 antibody) were added. IFN γ secretion was measured to evaluate the capability of DCs to prime the antigen specific T cells. The results showed that β -lap treatment greatly increased the IFN γ secretion and this effect was diminished when HMGB1 was neutralized or knocked down (Supporting figure R4c). Together, these data further confirmed that β -lap treatment induced dendritic cell-mediated T cell cross-priming and activation was in a tumor-derived HMGB1 dependent manner.

The figure (Fig.R4a and c) has been added to the Supplementary Fig. 7. In pages 10, lines 8-15 “HMGB1 was found to be critical for β -lap-induced immunogenicity in three experiments: (i) BMDCs were exposed to OVA protein for 4 hours, then naïve OT1 CD8⁺ T cells and the supernatants from β -lap-treated tumor cells (with/without α HMGB1 antibody or knockdown the HMGB1 in MC38 cells). IFN γ secretion was measured to evaluate the capability of BMDCs to prime the antigen specific T cells. The results showed that β -lap-treatment greatly increased the IFN γ secretion, and this effect was significantly diminished when HMGB1 was neutralized or knocked down (Supplementary Fig. 7a,b).” will be added to describe this.

Supporting figure R4: β -lap treatment induces dendritic cell-mediated T cell cross-priming and activation in a tumor-derived HMGB1 dependent manner. (a) HMGB1 expression in different MC38 cell lines (WT MC38, MC38 sheGFP and MC38 shHMGB1 cell pool) was determined by western blotting assay. (b) MC38 sheGFP or MC38 shHMGB1 cells were treated

with β -lap (4 μ M) for 3 hr followed by washing and replacing fresh medium, BMDCs with/without α HMGB1(20 μ g/ml) were cocultured with β -lap-treated tumor cells for another 24 hours. CD40 and CD86 expression on BMDCs were determined by flow cytometry analysis. (c) MC38 sheGFP or MC38 shHMGB1 cells were treated with β -lap (4 μ M) for 3 hr followed by replacing fresh medium for another 24 hours. BMDCs were exposed to 40 μ g/ml OVA protein for 4 hours, then naïve OT1 CD8⁺ T cells and the supernatants from β -lap-treated tumor cells (with/without α HMGB1 antibody) were added and allowed to incubate for another 48 hours. IFN- γ was determined by cytometric bead array assay.

Reviewer #2 (Remarks to the Author):

NQO1-based tumor targeting prodrug triggers innate sensing to overcome checkpoint blockade resistance. The authors have done a heroic amount of work in this manuscript. I have some specific questions that need addressing.

Comment: In general, I would like to see the cell number used in each *in vivo* tumor model (s.c) displayed at the top of each figure. Some of the tumor growth curves are different from figure to figure using the same cell lines and this is sometimes due to injected cell number and would help the reader greatly.

Response: We appreciate the reviewer's suggestion. In the revised manuscript, we have added the inoculated cell numbers for each tumor model in the figure legend part of Fig.1-6.

Comment: Also, "Spaghetti plots" of each tumor grown in individual mice should be shown in the supplemental data so growth of each tumor can be visualized. Error bars are impressively small given the low number of mice used in each group as these lines are known to have sizable variation in B6 mice. Spaghetti plots would help here.

Response: Thanks for your suggestion. In the revised manuscript, the "Spaghetti plots" of each tumor growth has been added to the Supplementary Fig. 10. In pages 16, lines 20-21 "The "Spaghetti plots" of each tumor growth curve of individual mouse for *in vivo* study was shown in Supplementary Fig.10." will be added to describe this.

Comment: Fig. 1C - Are these MC38 "null" or 'knock-down" for NQO1? Western analysis should be complemented with NGS analysis since there is still b-Lap dependent death of MC38 cells.

Response: Thanks for your suggestion. In this study, NQO1 gene in MC38 cells was knocked out by CRISPR/Cas9 technology. The guide sequence 5'-TTGTGTTTCGGCCACAATATC-3' was cloned into pSpCas9 (BB)-2A-Puro plasmid containing a puromycin selection gene. After transfection, we selected the puromycin-resistant stable cell clones. The expression of NQO1 in

MC38 parent cell and NQO1-KO cells was detected by western blotting assay. As shown in **supplementary figure 1b**, we did not detect the expression of NQO1 in MC38 NQO1-KO clones. We further determined the genomic editing of NQO1 by sequencing in the MC38 and MC38 NQO1 KO#5 cells. The results indicated that two alleles of the target gene NQO1 had been edited, one was frame shift mutation, and the other was loss of 6 base pairs (2 amino acids) with no shift mutation (**Supporting figure R5**).

Mouse NQO1

	PAM	Target sequence	
Wild type	5'- GGTGTCCA	CGGGGACATGAACGTCATTCTCTGGCCGATT	CAGGTAGCTCCTTCCAGA (0)
1	5'- GGTGTCCA	CGG-----GAACGTCATTCTCTGGCCGATT	CAGGTAGCTCCTTCCAGA (-6)
2	5'- GGTGTCCA	CGG-GACATGAACGTCATTCTCTGGCCGATT	CAGGTAGCTCCTTCCAGA (-1)

Supporting figure R5: The DNA fragments containing the target sequence for NQO1 knockout were amplified by PCR from genomic DNA from MC38 sgNQO1 5# cells. Then the PCR product was purified and sent out for sequencing. Allele 1 lost 6 base pairs, 2 amino acids, but no shift mutation. Allele 2 lost 1 base pair which cause shift mutation.

Comment: Fig 2A,B - Why do MC38 grow so differently in WT Vehicle mice in Fig. 2A v 2B? i.e 300mm in A at d25 versus >200mm in B? I ask this because the error in B is very small, yet if you look at A there would be 50% error.....

Response: Thank the reviewer for pointing this out. In Figure 2A and 2B, 6×10^5 MC38 cells were subcutaneously transplanted into C57BL/6 WT mice (n=5). Tumor volumes were measured at least twice weekly and calculated as $0.5 \times \text{length} \times \text{width} \times \text{height}$ and shown as mean \pm SEM. "Spaghetti plots" of each tumor grown in mice from vehicle group was shown (Supporting figure R6). As the reviewer mentioned, there are some growth variation in the vehicle groups in Figure 2A and 2B. Although we implanted the same cell numbers of MC38 cell in these two independent experiments, the tumor volume of each individual mouse had some variation even in the same batch of mice. The reason might because of different batch of mice and individual difference such as the exact age and weight of mice. For the difference of error bar might because of the variation of individual mice. As suggested, we added the "Spaghetti plots" of each tumor grown for most in vivo study in the supplementary figures and these may be helpful for better data presentation.

Supporting figure R6: "Spaghetti plots" of each tumor grown in mice from vehicle group of Fig.2A and Fig.2B.

Comment: The MC38 growth in the β -Lap treated group between Fig 6C and 4C look very different. 4C looks like a cure whereas 6C requires anti-PD1L for a cure. Why is this?

Response: The reviewer reached an important point. We apologize for the confusion. In preclinical and clinical practice, impressive clinical responses for immunotherapy is achieved only by treating very small tumor and/or at the early stage before fully developed tumor microenvironment. However, well-established tumors may generate complicated immunosuppressive networks and are generally refractory to immunotherapy, and required combination with other therapies. In this study, we mimicked this clinical practice in MC38 model with different tumor stages and treatment start time (**Fig. 6 a-c**). Similarly as noted above, in **Fig. 6 a-c**, β -lap monotherapy could induce the tumor regression and rejection when treated at Day 7 with a relative small tumor mass (around 50 mm³); however, the therapeutic effect was largely impaired when treated at Day 12 with an established larger tumor mass (150-200 mm³). The advanced tumor was also refractory to immunotherapy (**Fig. 6c**). Next, we tested whether combined local β -lap treatment with anti-PD-L1 treatment could control the established large MC38 tumors (**Fig.6c**). The results showed that β -Lap could eradicate large established and checkpoint blockade refractory tumors by combination with anti-PD-L1 therapy (**Fig.6c**). So the difference of therapeutic effect in smaller and advanced tumor should be because of the different status of immunosuppressive tumor microenvironment.

Comment: Fig 6 E and F legends are swapped around. E is the survival plot.

Response: We apologize for the mistake of the figure sequence. In the revised manuscript, we have corrected the sequence of Fig. 6e and f.

Comment: What is the effect of β -Lap on metastasis? Spontaneous metastasis such as E0771-LMB, or 4T1?

Response: Thanks for your suggestions. We have tested the effect of β -lap on spontaneous metastasis with murine breast 4T1 tumor model which is NQO1 positive (Supplementary Fig.1a). Briefly, 4T1 tumor bearing BALB/c mice were locally treated with β -lap (15 mg/kg, i.t.) every other day for four times. The growth curve of primary tumor was measured twice a week. Five weeks after tumor inoculation, tumor bearing mice were euthanized and the lungs were excised. As shown, β -lap treatment greatly inhibited the primary tumor growth (Supporting Figure R7a). Moreover, compared with the vehicle treatment group, the number of pulmonary metastases was significantly reduced in the β -lap treatment group (Supporting Figure R7b, c). These data indicate that β -lap is very potent in limiting tumor metastasis.

Supporting Figure R7: Local β -lap treatment inhibits 4T1 tumor growth and limits the pulmonary metastasis. BALB/c mice bearing 4T1 tumor (n=5-6/group) were locally treated with vehicle or β -lap (15 mg/kg, i.t.) every other day for four times. (a) tumor growth was monitored twice a week. b Day 37 after tumor inoculation, lungs were isolated for metastasis detection. Whole lung pictures were shown (b, upper panel), single cell suspensions were prepared from 100mg lung tissues, 1/300 of total cells were plated into 6 well plates. 10 μ g/ml 6-

Thioguanine was added into the culture medium for 4T1 cell selection. 5 days later, crystal violet staining was used to detect the 4T1 clones (b, lower panel). Number of the 4T1 clones was counted (c). Data is shown as mean \pm SEM, **P < 0.01, unpaired student t-test was used to analyze the significance of changes.

Reviewers' comments:

Reviewer #1 (Remarks to the Author):

The authors have addressed many of the comments, but the critical experiment has not been done.

Comment 2 asked for clarification that β -lapachone caused cell death in NQO1-deficient B16 cells in Figure 1a, but not cell death in Figure 1d. The authors explained that the high-concentration β -lapachone had an "off target" effect and revised the manuscript by deleting the 8 μ M treatment group in Figure 1a. This reviewer thinks that it is necessary to re-exam Figure 1a and d under the same conditions.

The authors obtained data using mainly NQO1-expressing MC38 and TC-1 cells in the mechanism studies. Comment 4 asked for the analysis of β -lapachone-induced innate sensing "in NQO1-deficient cells" using overexpression of NQO1 in the mechanism studies in Figure 2-6. Therefore, the experiments are needed to clarify the data.

Comment 5 asked for an analysis of the infiltration of T cells using immunohistochemistry. The authors demonstrated CD8+ T cell infiltration in combination with β -lapachone monotherapy or anti-PD-L1 therapy using immunofluorescence in Figure R2. Rather, in the Figure R2, this reviewer confuses a correlation between inhibition of PD-L1 and infiltration of cytotoxic T cells.

Therefore, this reviewer feels that the authors need to do a lot more experimental work to consolidate their findings.

Response to reviewer:

Reviewer #1:

1. Comment 2 asked for clarification that β -lapachone caused cell death in NQO1-deficient B16 cells in Figure 1a, but not cell death in Figure 1d. The authors explained that the high-concentration β -lapachone had an “off target” effect and revised the manuscript by deleting the 8 μ M treatment group in Figure 1a. This reviewer thinks that it is necessary to re-exam Figure 1a and d under the same conditions.

Response: We appreciated the reviewer’s suggestion. In the revised manuscript, we re-exam the **Figure 1a** and **Figure 1d** under the same conditions. Briefly, 4000 cells (per well) in 96-well plates were exposed to β -lap with or without NQO1 inhibitor DIC for 3 hr and cell survival was assessed 48 hr later with SRB cytotoxicity assay. The results showed that tumor cell lines (MC38, TC-1 and Ag104Ld) that express high level of NQO1 were sensitive to β -lap exposure with a lethal dose around 2-4 μ M. In contrast, NQO1 deficient cell lines, B16 and Panc02 were resistant to β -lap exposure. Overexpression of NQO1 in B16 cells led to sensitivity to β -lap, and inhibition of NQO1 by dicoumarol spared β -lap lethality (Fig. 1d). **Figure 1a** and **Figure 1d** have been replaced with new data.

2. The authors obtained data using mainly NQO1-expressing MC38 and TC-1 cells in the mechanism studies. Comment 4 asked for the analysis of β -lapachone-induced innate sensing “in NQO1-deficient cells” using overexpression of NQO1 in the mechanism studies in Figure 2-6. Therefore, the experiments are needed to clarify the data.

Response: We appreciated the reviewer’s suggestion. In our study, we have proved that the β -lapachone could induce innate sensing and antitumor immune response in NQO1 highly expressing MC38 tumor models. In this revised manuscript, we have some additional data to confirm that β -lapachone could trigger innate sensing in NQO1 overexpressed B16 models. Firstly, we carried out direct killing assay by *in vitro* assays in NQO-null and NQO1-overexpressing B16 cells. B16 (NQO1-null) was resistant to β -lap exposure, and overexpression of NQO1 in B16 cells led to sensitivity to β -lap (Figure R1a and Figure 1d). Inhibition of NQO1

by dicoumarol spared β -lap lethality in NQO1-overexpressing B16 cells (Figure R1a and Figure 1d). Next, we examined the antitumor efficacy of β -lap in NQO-null and NQO1-overexpressing B16 cells. NQO1-null B16 tumors didn't respond to β -lap treatment (Figure R1b and Supplementary Fig. 1k). In sharp contrast, NQO1 overexpressing (clone #1 and #4) tumor bearing mice showed a dramatic tumor suppression after β -lap treatment (Figure R1b and Supplementary Fig. 1k). To determine whether β -lapachone-induced innate sensing was response for the antitumor efficacy of β -lap in NQO1-overexpressing B16 cells, we analyzed the innate danger protein HMGB1 secretion in β -lap treated tumor cells *in vitro*. Indeed, like in MC38 cells, we observed a dose-dependent secretion of HMGB1 in NQO1-overexpressing B16 cells but not in NQO1-null B16 (Figure R1c and Figure 5a). To determine whether β -lapachone induced innate sensing and activated type I IFNs in B16NQO1 tumor cells and whether this effect was dependent on HMGB1/TLR4/MyD88 pathway, we co-cultured β -lapachone treated B16 NQO1 cells and BMDCs from WT or *MyD88*^{-/-} mice. Indeed, BMDCs greatly increased the production of IFN β protein after co-culture, and MyD88 deficiency completely abolished the IFN β production (Figure R1d and Supplementary Fig. 5c). To prove the β -lapachone induced innate sensing and type I IFNs' production were required for the antitumor immune response *in vivo*, we inoculated NQO1-overexpressing B16 cells separately in WT mice, immune deficient mice (*Rag1*^{-/-}), MyD88 deficient mice (*MyD88*^{-/-}) and type I IFNs receptor deficient mice (*Ifnar1*^{-/-}), and treated with β -lapachone. The result indicated that β -lapachone dramatically suppressed the B16 NQO1 growth (Figure R1e and Supplementary Fig. 2a), and lost therapeutic activity in immune-deficient *Rag1*^{-/-} mice (Figure R1f and Supplementary Fig. 2a), MyD88 deficient mice (Figure R1g and Supplementary Fig. 6b) and type I IFNs receptor deficient mice (Figure R1h and Supplementary Fig. 6c). All these results above were consistent with that from MC38 tumor model. Moreover, we further determined where β -Lap eradicates checkpoint blockade refractory B16 NQO1 tumors by combination with anti-PD-L1 therapy. NQO1 overexpressing B16 tumors failed to respond to anti-PD-L1 Ab alone. By contrast, β -lap monotherapy largely inhibited the growth of the B16-NQO1 tumors. Strikingly, when combined with PD-L1 blockade, β -lap had a markedly synergetic antitumor effect (Figure R1e and 1j and Supplementary Fig. 8f). Similar synergetic effect was also observed in MC38 models.

Sentences that describe these data, have been highlighted in the manuscript.

Supporting figure R1: β -Lap induced innate sensing and antitumor immune response in NQO1-expressing B16 tumor models. (a) B16 cells and NQO1 overexpressing B16 cells were planted in 96 well plates were exposed to β -lap \pm dicoumarol (DIC, 50 μ M) for 3 hr pulse and survival assessed 48 h later. (b) C57BL/6 mice (n=4/group) were transplanted with parental B16 cells (NQO1 null) or NQO1 stable overexpressing B16 cells (clone #1 and #4) and treated with

β -lap every other day for four times. (c) B16 cells (NQO null and overexpression clones) were treated with β -lap for 3 hr followed by washing and replacing medium. The level of HMGB1 released into the culture supernatant was determined by ELISA 24 hr later. (d) NQO1-overexpressing B16 cells (B16NQO1#1) were treated with β -lap (4 μ M) for 3 hr followed by washing and replacing fresh medium. 24 hr later, BMDCs from WT or *Myd88*^{-/-} mice were cocultured with β -lap-treated tumor cells for another 48 hr. The level of IFN β from the culture supernatant was detected by ELISA. (e-h) NQO1 overexpressing B16 cells (B16 NQO1 #1) were subcutaneously transplanted into C57BL/6 WT (e), *Rag1*^{-/-} (f), *Myd88*^{-/-} (g), and *Ifnar1*^{-/-} mice (n=4-5/group), respectively. Tumor bearing mice were treated with β -lap every other day for four times. (i, j) B16 cells with stable NQO1 overexpression (Mixed clone #1, #3 and #4) were s.c. inoculated into C57BL/6 mice. Tumor bearing mice (about 100 mm³) were treated with β -lap for four times with or without anti-PD-L1 based checkpoint blockage for three times.

3. Comment 5 asked for an analysis of the infiltration of T cells using immunohistochemistry. The authors demonstrated CD8⁺ T cell infiltration in combination with β -lapachone monotherapy or anti-PD-L1 therapy using immunofluorescence in Figure R2. Rather, in the Figure R2, this reviewer confuses a correlation between inhibition of PD-L1 and infiltration of cytotoxic T cells.

Response: Thank the reviewer for raising this question and sorry for the confusing. PD-1 is the central inhibitory receptor in regulating CD8 T cell exhaustion during cancers¹⁻⁴, and PD1 is always upregulated on tumor-infiltrating T cells^{1,5,6}. PD-L1 expression on both tumor cells and host cells have been reported to impair T-cell proliferation and effector function to mediate immunoevasion^{2,7,8}. PD-1/PD-L1 blockade has been demonstrated to increase the proliferation and function of tumor infiltrating CD8 T cells, and encourage the antitumor effects in several solid tumors^{2,9,10}. Our immunofluorescence staining data that anti-PD-L1 monotherapy can increase the number of tumor infiltration CD8 T cells, is consistent with previous studies, and β -lapachone combination treatment dramatically magnify this effect (Supplementary Fig.9). We have clarified this argument in our revised manuscript. In page 14, lines 27-29 “Encouragingly, PD-1/PD-L1 blockade has shown promising capacity to increase the proliferation and function of tumor infiltrating CD8⁺ T cells, and enhance the antitumor efficacy in several cancer types.” has been added to describe this.

Reference

- 1 Ahmadzadeh, M. *et al.* Tumor antigen-specific CD8 T cells infiltrating the tumor express high levels of PD-1 and are functionally impaired. *Blood* **114**, 1537-1544, doi:10.1182/blood-2008-12-195792 (2009).
- 2 Dong, H. *et al.* Tumor-associated B7-H1 promotes T-cell apoptosis: a potential mechanism of immune evasion. *Nat Med* **8**, 793-800, doi:10.1038/nm730 (2002).
- 3 Hashimoto, M. *et al.* CD8 T Cell Exhaustion in Chronic Infection and Cancer: Opportunities for Interventions. *Annu Rev Med* **69**, 301-318, doi:10.1146/annurev-med-012017-043208 (2018).
- 4 Pardoll, D. M. The blockade of immune checkpoints in cancer immunotherapy. *Nat Rev Cancer* **12**, 252-264, doi:10.1038/nrc3239 (2012).
- 5 Baitsch, L. *et al.* Exhaustion of tumor-specific CD8(+) T cells in metastases from melanoma patients. *J Clin Invest* **121**, 2350-2360, doi:10.1172/Jci46102 (2011).
- 6 Fourcade, J. *et al.* Upregulation of Tim-3 and PD-1 expression is associated with tumor antigen-specific CD8(+) T cell dysfunction in melanoma patients. *J Exp Med* **207**, 2175-2186, doi:10.1084/jem.20100637 (2010).
- 7 Juneja, V. R. *et al.* PD-L1 on tumor cells is sufficient for immune evasion in immunogenic tumors and inhibits CD8 T cell cytotoxicity. *J Exp Med* **214**, 895-904, doi:10.1084/jem.20160801 (2017).
- 8 Tang, H. D. *et al.* PD-L1 on host cells is essential for PD-L1 blockade-mediated tumor regression. *J Clin Invest* **128**, 580-588, doi:10.1172/Jci96061 (2018).
- 9 Brahmer, J. R. *et al.* Safety and activity of anti-PD-L1 antibody in patients with advanced cancer. *N Engl J Med* **366**, 2455-2465, doi:10.1056/NEJMoa1200694 (2012).
- 10 Topalian, S. L. *et al.* Safety, activity, and immune correlates of anti-PD-1 antibody in cancer. *N Engl J Med* **366**, 2443-2454, doi:10.1056/NEJMoa1200690 (2012).

REVIEWERS' COMMENTS:

Reviewer #1 (Remarks to the Author):

This reviewer appreciates the extra effort of the authors to address the concerns raised.

The authors have properly addressed most of the requirements and suggestions.